# Non-covalent inhibitors of thioredoxin glutathione reductase with schistosomicidal activity in vivo

Valentina Z. Petukhova[1,8], Sammy Y. Aboagye[2,8], Matteo Ardini[3,8], Rachel P. Lullo[2], Francesca Fata[3], Margaret E. Byrne[2], Federica Gabriele[3], Lucy M. Martin[2], Luke N. M. Harding[1], Vamshikrishna Gone[4], Bikash Dangi[5], Daniel D. Lantvit[4], Dejan Nikolic[1], Rodolfo Ippoliti[3], Grégory Effantin[6], Wai Li Ling[6], Jeremy J. Johnson[5], Gregory R. J. Thatcher[7], Francesco Angelucci[3] ✉, David L. Williams[2] ✉ & Pavel A. Petukhov[1] ✉

Only praziquantel is available for treating schistosomiasis, a disease affecting more than 200 million people. Praziquantel-resistant worms have been selected for in the lab and low cure rates from mass drug administration programs suggest that resistance is evolving in the field. Thioredoxin glutathione reductase (TGR) is essential for schistosome survival and a validated drug target. TGR inhibitors identified to date are irreversible and/or covalent inhibitors with unacceptable off-target effects. In this work, we identify non-covalent TGR inhibitors with efficacy against schistosome infections in mice, meeting the criteria for lead progression indicated by WHO. Comparisons with previous in vivo studies with praziquantel suggests that these inhibitors outperform the drug of choice for schistosomiasis against juvenile worms.

Schistosomiasis is a devastating but neglected tropical disease with more than 200 million people infected resulting in more than 200,000 deaths annually[1,2]. In addition, almost everyone infected has a significant degree of disability[3]. Female genital schistosomiasis is a common complication, occurring in approximately 40 million girls and women, making it one of the most common gynecologic conditions in Africa, and schistosome infections have been implicated as cofactors in the acquisition and transmission of HIV and are a WHO-recognized risk factor for HIV infection[4,5]. Schistosomiasis control strategies rely almost exclusively on mass drug administration (MDA) using praziquantel (PZQ) monotherapy. No alternatives to PZQ are currently available and few drugs or vaccines are in the clinical pipeline for schistosomiasis treatment[6–8]. Given this situation it is particularly concerning that MDA campaigns show that PZQ cure rates are often less than 50% and modeling studies demonstrating that these campaigns with PZQ alone are unlikely to interrupt transmission, and once MDA is suspended, the prevalence of infection is likely to rebound to pre-MDA levels[9–11]. Furthermore, with large-scale drug use it is inevitable that PZQ-resistant parasites will evolve, and PZQ resistance has been induced in laboratory infections[12]. In addition, PZQ has limited activity against juvenile liver-stage worms[13] and, although progress is being made[14], it is difficult to administer to children. Therefore, the identification of new drugs for this disease is indispensable.

We have previously shown that schistosome redox defenses are limited, and that as such they can be disrupted pharmacologically leading to worm death in vitro and in vivo[15]. Additionally, we have found that schistosomes do not have either an authentic glutathione

[1]Department of Pharmaceutical Sciences, College of Pharmacy, University of Illinois at Chicago, Chicago, IL, USA. [2]Department of Microbial Pathogens and Immunity, Rush University Medical Center, Chicago, IL, USA. [3]Department of Life, Health and Environmental Sciences, University of L'Aquila, L'Aquila, Italy. [4]UICentre, Department of Pharmaceutical Sciences, College of Pharmacy, University of Illinois at Chicago, Chicago, IL, USA. [5]Department of Pharmacy Practice, College of Pharmacy, University of Illinois at Chicago, Chicago, IL, USA. [6]University of Grenoble Alpes, CEA, CNRS, IBS, F-38000 Grenoble, France. [7]Department of Pharmacology & Toxicology, R. Ken Coit College of Pharmacy, University of Arizona, Tucson, AZ, USA. [8]These authors contributed equally: Valentina Z. Petukhova, Sammy Y. Aboagye, Matteo Ardini. ✉e-mail: francesco.angelucci@univaq.it; david_williams@rush.edu; pap4@uic.edu

disulfide reductase (GR) or thioredoxin reductase (TrxR) enzymes. Instead, both these activities are provided by selenocysteine (Sec/U)-containing thioredoxin glutathione reductase (TGR)[16]. TGR is a 130 kDa obligate homodimer as the functional stereochemistry of the FAD redox site in each subunit is generated by protein dimerization. The enzymatic cycle of TGR (Fig. 1a) can be subdivided into a reductive half-reaction (electrons flow from NADPH to the enzyme) and an oxidative half-reaction (electrons from the enzyme are transferred to the oxidizing substrates)[17]. Reducing equivalents from NADPH are transferred to the FAD redox site of TGR comprised by the isoalloxazine ring of the flavin and a pair of nearby cysteines (C154/C159). From here, electrons are transferred to the C-terminal redox site of the other subunit where the Sec residue lies adjacent to another cysteine (C597'/U598'). This mobile redox active arm can transfer electrons to the incoming oxidized Trx or internally to the Grx domain (C28/C31) of the partner subunit where reduction of oxidized glutathione (GSSG) occurs. The TGR species with two electrons accepted from one NADPH is the $EH_2$ form. Following generation of the $EH_2$ a second NADPH binds and donates electrons generating the $EH_4$ form, which is competent for substrate reduction. Using both reverse genetic and pharmacological approaches, we have demonstrated that TGR is an essential and druggable target for the treatment of schistosomiasis[18–20].

Further development of TGR-based therapeutic approaches is impeded by the lack of non-covalent small molecule inhibitors of TGR, a well-known and general problem for all the members of the high molecular weight, Sec-containing TrxR subfamily that are considered attractive and unexploited drug targets for several human diseases[21,22]. Known inhibitors of TGRs/TrxRs, which include metal complexes and compounds containing Michael acceptors or oxadiazole oxides[19,23,24], usually react with Sec and Cys residues and other biogenic nucleophiles irreversibly. Covalent/irreversible inhibitors target the redox/catalytic apparatus of TGR with almost no recognition elements that give affinity and selectivity for TGR over off-targets and the cellular redox apparatus resulting in undesirable cross-reactions and low specificity in vivo[25]. Although anti-infective covalent inhibitors can offer long residence times in their targets and low systemic circulation time in the host (see ref. 26 and the references therein), they have limitations associated with well-known multiple adverse reactions, genotoxicity, and special requirements for storage, handling, and administration, posing challenges with respect to the standard operating procedure (SOP) for MDA campaigns requiring single or double dose activity, oral administration, minimal toxicity, and stability in tropical settings[27]. Both potassium antimonial tartrate and oltipraz inhibit TGR[18] by a covalent mechanism and were used clinically for schistosomiasis treatment but were discontinued due to unacceptable side effects[28,29].

Our extensive biochemical analyses of TGR combined with 3D structures of TGR complexes with various substrates, inhibitors, and in different redox states[17,30–32], allow us to define in detail the catalytic mechanisms of TGR. In our recent studies, we have identified a different inhibition mechanism of TGR in which an inhibitor binds non-covalently at the so-called doorstop pocket interfering with enzyme function[33,34].

In this work, small molecule fragments obtained by X-ray crystallography are ligated and partially optimized as non-covalent, and orally bioavailable inhibitors of TGR. We report the cryo-EM structure of TGR demonstrating that these inhibitors bind at the doorstop pocket preventing the NADPH oxidation steps. These compounds display schistosomicidal activity against different parasite-life cycle stages reaching the nanomolar range. Most importantly, they demonstrate efficacy against schistosome infections in mice, meeting the criteria for lead progression indicated by WHO, and outperform the previously reported efficacy of praziquantel against juvenile worms[13].

## Results

### Fragment-based drug design and chemistry

Utilizing "actives" identified in a quantitative high-throughput screen against *Schistosoma mansoni* TGR[20,35], 92 commercially available low molecular weight (MW) compounds/fragments were tested by X-ray crystallography[33,34,36]. Some of the fragments, for which X-ray structures were obtained, were found in the doorstop pocket expected to be critical for TGR inhibition and adjacent to the NADPH binding site (Fig. 1b, c)[33]. In the doorstop pocket, subpocket A binds 2-carboxynaphthyridine and subpockets B and C bind 4-(2-hydroxyethyl)−1-piperazineethane (HEPE) as shown in Fig. 1c. TGR $IC_{50}$s for these low MW fragments varied from 0.76 mM to 4.4 mM. Considering their low MW and non-covalent nature of inhibition of TGR, these compounds were deemed appropriate for fragment-based design.

Further structure-based iterative optimization resulting in inhibitors shown in Fig. 1d was driven by bioisosteric replacement of the fragments bound to subpockets A-C, scaffold-hopping, de novo design, and medicinal chemistry and was facilitated by SZMAP/GamePlan[37], vBrood[38], and MOE software[39]. The computational analysis in Fig. 1e, f was facilitated by combining two X-ray fragments found in subpockets A-C in PDB:6FP4 and PBD:6FTC via a short $CH_2CH_2$ linker in a putative chimera molecule. Since all the fragments crystallized in subpocket C contained HEPE moiety with an additional polar substituent, it was tempting to think that a similar polar moiety should be placed in subpocket C. We noticed, however, that the bottom part of subpocket C in the vicinity of the fragments found in the X-ray structures is formed by the hydrophobic portions of D325, Y479, A481, H538, G483, V469, T471 and is likely to prefer bulky hydrophobic substituents. Consistent with this observation, all non-polar attachments proposed by GamePlan in subpocket C congregated near the piperazine ring of HEPE at the bottom of subpocket C (Fig. 1e). The published TGR X-ray structures also contained several water molecules in subpockets A and C (Fig. 1f). Considering that the water molecules were trapped largely in the hydrophobic pockets, often made contacts with or were displaced by the fragments, and did not appear to play a structural role, we anticipated that their displacement with larger non-polar moieties would improve binding of newly designed inhibitors. An analysis of the free energy minima and maxima for the water probe in subpockets A-C using SZMAP[37] and WaterOrientation extension in VIDA[40] is shown in Fig. 1f and Supplementary Movie 1. The bottom of subpocket C in proximity to the chimera ligand contained several mostly non-polar (purple) probes with positive free energy values, indicating that it may be filled with additional non-polar substituents. The locations of waters found in the X-ray structures either overlapped or were very close to the locations of the probes placed by SZMAP. This observation further strengthened the notion that additional binding free energy could be gained not only by forming meaningful interactions with the doorstop pocket but also by liberating the trapped waters.

Further visual inspection in VIDA of the vBrood search for candidates that can fill the doorstop pocket led us, among other possible scaffolds, to pinane in subpocket C. The $sp^3$ carbon scaffold of the pinane rings offered an effective way to capture interactions with the hydrophobic portions of subpocket C while also displacing likely trapped waters. Subpockets A and B are also mostly hydrophobic and with only a few polar groups available to make polar interactions with the ligand. While subpockets B and C are rather large, subpocket A is narrow and restricted in depth by FAD at the bottom of the pocket, limiting the choice of modifications of the inhibitor in subpocket A. Although the SZMAP and GamePlan analyses (Fig. 1e, f) suggested placement of polar and non-polar substituents in multiple locations, including those that were relatively remote, we intentionally limited the modifications to those in proximity or to the putative chimera molecule itself to maximize the potential for further improvements during structure-activity relationship (SAR) studies. By advancing

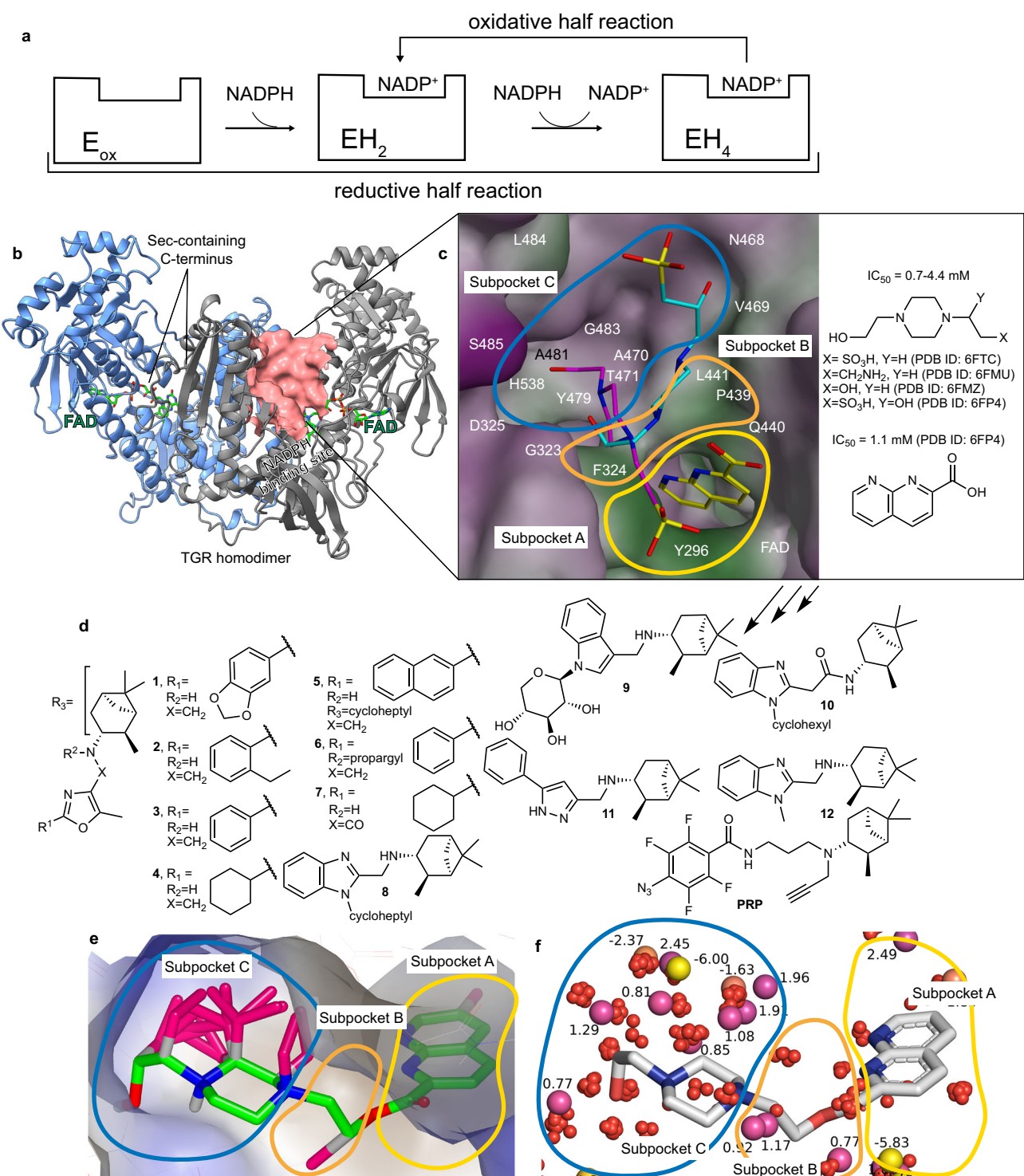

**Fig. 1 | From small molecules fragments bound to the doorstop pocket to the inhibitors designed in this study. a** A simplified picture of the enzymatic mechanism of TGR is shown (for a comprehensive TGR mechanism see refs. 17, 30). In the reductive half-reaction 2 eq of NADPH are consumed to produce the EH$_4$ species, the 4-electron state, the one competent for substrate reduction in the oxidized half-reaction. After the initial reduction of the oxidized enzyme (E$_{ox}$) to EH$_2$, the 2-electron reduced state, TGR oscillates between EH$_2$ and EH$_4$ during turnover. **b** TGR homodimer is shown in cartoons and each subunit is differently colored. The FAD cofactor is in green sticks. The doorstop pocket adjacent to the NADPH binding site is shown as a pink surface in one subunit. **c** A magnification of the doorstop pocket with representative bound fragments identified by X-ray crystallography (PDB ID 6FTC – magenta and PDB ID 6FP4 – cyan and yellow models); the molecular surface of the doorstop pocket is colored according to its

hydrophobic features (green = hydrophobic; magenta = hydrophilic). Subpockets A-C are outlined in different colors and for each fragment the PDB ID is reported. **d** The TGR inhibitors designed in this study. **e** Gameplan non-polar hypotheses (magenta sticks) generated for chimera molecule (green) built using the X-ray fragments found in subpockets A-C in PDB ID 6FP4 and PBD ID 6FTC and connected by a short CH$_2$CH$_2$ linker to facilitate calculations and analysis. The binding site surface is colored with VIDA hydrophobicity palette, brown is hydrophobic, and blue is hydrophilic. **f** SZMAP grid results for TGR-chimera (gray sticks) complex processed with WaterOrientation VIDA extension show the most probable probe positions (polar substituents - yellow bubbles, non-polar substituents - purple bubbles) and the corresponding free energy values and the location of the water molecules (red bubbles) found in the X-ray structures of TGR.

**Table 1 | Biochemical characterization of 1–10, controls 11, 12, auranofin (AF), praziquantel (PZQ), meclonazepam (MZM) and photoreactive probe (PRP)**

| | ID | *S. mansoni* TGR IC$_{50}$ (µM)[a] | *S. mansoni* TGR inhibition (%) at 66.7 µM[b] | Human TrxR1 IC$_{50}$ (µM)[a] | Human TrxR1 IC$_{50}$ (µM) or inhibition (%) at 66.7 µM[b] | Human GR (inhibition at 66.7 µM) | Decrease (%) in GSH/GSSG ratio[c] |
|---|---|---|---|---|---|---|---|
| **Slow TGR inhibitors** | 1 | >66.7 | 42.1% | >66.7 | 37.0 ± 3.44 | n.i. | 75 |
| | 2 | >66.7 | 28.7% | >66.7 | 50.0 ± 13.6 | n.i. | 39 |
| | 3 | >66.7 | 45.0% | >66.7 | 42.7 ± 3.95 | n.i. | 62 |
| | 4 | >66.7 | 68.6% | >66.7 | 19.0 ± 5.72 | n.i. | 60 |
| | 5 | >66.7 | 73.0% | >66.7 | 28.5% | n.i. | 56 |
| **Fast TGR inhibitors** | 6 | 2.5 ± 0.51 | — | >66.7 | 28.9% | n.i. | 32 |
| | 7 | 14.6 ± 1.22 | — | 22.3 ± 2.03 | — | n.i. | 38 |
| | 8 | 10.3 ± 2.01 | — | 4.9 ± 1.72 | — | n.i. | 20 |
| | 9 | 57.5 ± 2.46 | — | 70.5 ± 11.4 | — | n.i. | 9 |
| | 10 | 18.6 ± 2.48 | — | 13.4 ± 2.0 | — | n.i. | n.d.[d] |
| **Control** | 11 | n.i.[e] | n.i.[e] | n. d.[d] | n.d.[d] | n.i. | 0.4 |
| | 12 | n.i.[e] | n.i.[e] | >66.7 | 28.6% | n.i. | 1.7 |
| | PRP[f] | 34.9 ± 6.11 | — | n.d.[d] | n.d.[d] | n.d.[d] | n.d.[d] |
| | AF | 0.007 | — | 0.02 | — | n.d.[d] | 91 |
| | PZQ | — | n.i.[e] | — | n.i.[e] | n.d.[d] | 1 |
| | MZM | — | n.i.[e] | — | n.i.[e] | n.d.[d] | 1 |

Data are represented by $n$ = 3 independent experiments as mean ± SD. Source data are provided as a Source Data file.

[a]IC$_{50}$ after 15 min preincubation (enzyme + 100 µM NADPH + compound).

[b]IC$_{50}$ (µM) or inhibition (%) at 66.7 µM (enzyme + 100 µM NADPH + compound) after 6 h preincubation. Inhibition (%) of TGR activity after 6 h incubation are given and not IC$_{50}$s because equilibrium between enzyme and inhibitor was not obtained in this time frame (Fig. 3d).

[c]ratio of GSH/GSSG in adult worms determined after 3 h exposure to compounds at 50 µM.

[d]*n.d.* not determined.

[e]*n.i.* no inhibition at 67 µM in 6 h.

[f]PRP is a "slow" inhibitor.

through several generations of inhibitors, a series of inhibitors of TGR was obtained (Fig. 1d) with activity against recombinant TGR improving from the mM range for the fragments alone to the single digit µM range. Of more than 100 compounds synthesized and tested for TGR inhibition (a more thorough description of compound design will form the basis of a future manuscript), the most promising were selected to be extensively characterized in the in vitro and in vivo studies described below. The candidates selected are predicted to be orally bioavailable according to the analysis of Lipinski et al.[41] (Table 1, Supplementary Table 1). The synthesis and characterization of compounds in Table 1 are described in Supplementary Methods, Supplementary Figs. 1–20, and Supplementary Tables 1–14.

## TGR is inhibited in *S. mansoni* worms

The activities against recombinant TGR of the inhibitors in Fig. 1d ranged from a modest 28.7% inhibition at 67 µM to robust activity with IC$_{50}$ = 2.5 µM (Table 1). A predicted outcome of TGR inhibition in worms is the accumulation of oxidized GSH (GSSG) because of attenuated reduction to GSH by TGR, leading to decreases in the GSH/GSSG ratio. To characterize compound engagement of TGR in adult worms ex vivo, the GSH/GSSG ratio was measured for compounds in Table 1. After a 3 h exposure to compounds at 50 µM, large decreases in GSH/GSSG of 20 to 75% were observed. Under the same conditions, treatment with the current drug of choice, PZQ, or the clinically tested and now discontinued meclonazepam (MZM)[42], which have different schistosomicidal mechanisms and do not inhibit TGR, resulted in no change of that ratio. Treatment with positive control auranofin (AF), a covalent TGR inhibitor with schistosomicidal activity[18], decreased GSH/GSSG ratio by 90%. Treatment with inactive, negative control compounds (11 and 12), structurally related to the active inhibitors 1-10, had minimal effect on the GSH/GSSG in treated worms.

To further characterize compound engagement of TGR in ex vivo worms using an orthogonal assay, inhibition of TGR activity in newly transformed schistosomula (NTS, skin-stage worms) was assessed using a TrxR-selective fluorescent probe[43,44], TRFS-Green. The fluorescence of TRFS-Green is induced by the TGR (or TrxR)-mediated disulfide cleavage followed by intramolecular cyclization to liberate the masked naphthalimide fluorophore. Treatment of NTS with inhibitors 1, 2, 4, 7, and 8, positive control AF, or negative control 12 for 2 h was followed by addition of TRFS-Green. Fluorescence in wells was determined hourly for the first 5 h and at the terminal point of 24 h after addition of probe. NTS treated with TRFS-Green only were clearly fluorescent after 2 h incubation (Fig. 2, Supplementary Fig. 21). Treatment with inhibitors 1, 2, 4, 7, and 8, and AF led to significant decreases in fluorescence. Consistent with the outcome of the measurements of the GSH/GSSG ratio, negative control 12 had negligible effects on TRFS-Green fluorescence (Fig. 2). Overall, these findings indicate that the TGR inhibitors engage TGR in ex vivo worms.

## TGR inhibitors do not react covalently with GSH, selenocysteine, or TGR, and inhibition of TGR is reversible

To evaluate stability of the TGR inhibitors in the presence of GSH and selenocysteine, compound 2 and a known covalent inhibitor of TGR, TRi-1[20,45], were incubated with GSH or the N-Boc protected methyl ester of selenocysteine, and the reaction mixtures were analyzed by LCMS (Supplementary Figs. 22 and 23). Unlike TRi-1, no reaction of 2 with either thiol or selenol groups in GSH or protected selenocysteine was observed. Incubation of inhibitor 2 under the biochemical assay conditions with TGR and with or without NADPH resulted in no formation of derivatives of 2. To evaluate reversibility, TGR inhibitors were tested in the jump dilution assay[46,47]. In the jump dilution assay, after incubation of enzyme, NADPH, and inhibitor, the reactions are diluted to well below the IC$_{50}$ for the inhibitor allowing its release from the enzyme and activity is determined. We find that our inhibitors are reversible, while known covalent TGR inhibitors[20,45], including AF, Stattic, and TRi-1, are irreversible (Fig. 3a).

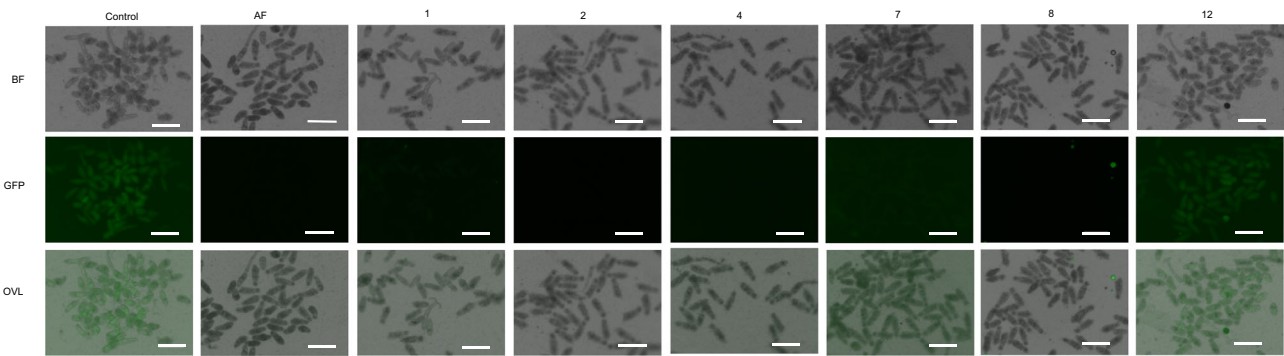

**Fig. 2 | Inhibition of TGR in newly transformed schistosomula (NTS) visualized with TRFS-Green.** Representative images from two independent experiments in bright field (BF), green fluorescent protein filter (GFP, $\lambda_{EX} = 438$ nm, $\lambda_{EM} = 538$ nm), and overlay (OVL) of newly transformed schistosomula (NTS) after 2 hr culture with inhibitor (AF @ 3 μM, other compounds @ 30 μM) followed by addition of TRFS-Green (10 μM) for 4 hr. Source data are provided as a Source Data file. Scale bar = 200 μm.

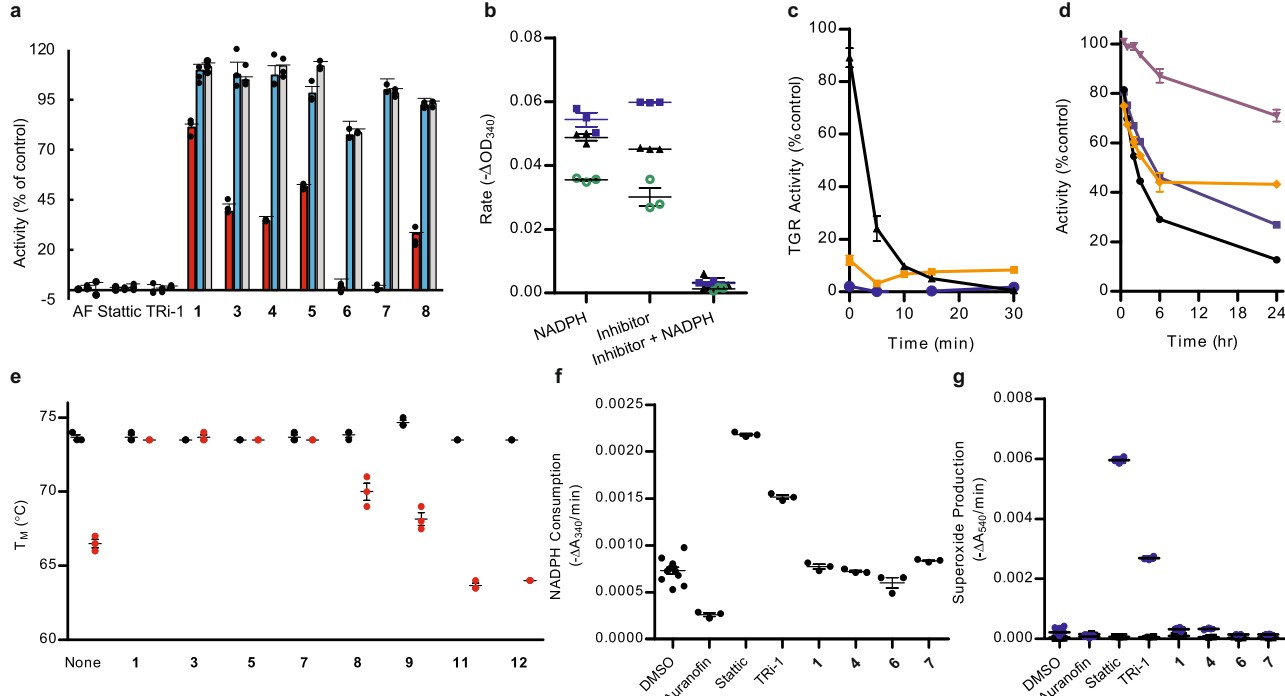

**Fig. 3 | Functional characterization of the inhibitors. a** Reversibility of TGR inhibitors. Activity (5,5′-dithiobis (2-nitrobenzoic acid) (DTNB) reduction) of the enzyme (3.7 nM) incubated with 250 μM inhibitor for 15 min was determined (red bars). TGR (370 nM) was incubated with 250 μM compound and 100 μM NADPH for 15 min. The sample was diluted 100-fold and the activity was determined immediately (blue bars) and after 60 min (light grey bars). AF = auranofin. Average ± standard deviation (n = 3) shown. **b** NADPH dependence of inhibitors. Activity of TGR after exposure to **6** (green circles), **7** (blue squares), or **8** (black triangles) with or without initial incubation with NADPH compared to control TGR incubated with NADPH and without inhibitor. Average ± standard deviation (n = 3) shown. **c** Time dependence of inhibition. Inhibition of DTNB reduction by TGR in the presence of 50 μM fast inhibitors **6** (orange), **7** (blue), **8** (black) compared to TGR incubated without compound. Average ± standard deviation (n = 3) shown. **d** Time dependence of inhibition. Time-dependent activity (DTNB reduction) of TGR in the presence of 50 μM slow inhibitors **1** (purple), **3** (orange), **4** (black), **5** (blue) compared to TGR activity incubated in the absence of compounds. Average ± standard deviation (n = 3) shown. **e** Compound effect on thermal stability of TGR. Melting temperature without (black) or with addition of 500 μM NADPH (red) of TGR alone or with 100 μM inhibitor **1, 3, 5, 7, 9** or control compounds **11** or **12**. Average ± standard deviation (n = 3) shown. **f** Oxidase activity after incubation with inhibitors. Consumption of NADPH ($\Delta A_{340}$/min) by TGR after exposure to inhibitors for 15 min in presence of NADPH. Average ± standard deviation (n = 3) shown except control n = 12. **g** Production of superoxide by TGR after incubation with inhibitors. Superoxide production was determined by measuring consumption of pyrogallol red ($\Delta A_{540}$/min) without added superoxide dismutase (blue circles) and with added superoxide dismutase (black squares) Average ± standard deviation, n = 3 except Auranofin n = 2, TRi-1 with SOD, n = 2, compound **4** without SOD, n = 2, and compound **6** with SOD, n = 2. Source data are provided as a Source Data file.

## TGR inhibitors target a reduced form of TGR

Next, we investigated if the TGR redox state affects the binding and activity of the inhibitors. Inhibitors **6-8** were incubated with TGR with and without NADPH, and the inhibition of TGR activity was determined after adding a second aliquot of NADPH and 5,5′-dithiobis (2-nitrobenzoic acid) (DTNB). Incubation of these inhibitors with TGR in the absence of NADPH led to significantly attenuated enzyme inhibition compared to inhibition resulting from incubation in the presence of NADPH (Fig. 3b).

Typical TGR inhibitor studies utilize a 15-minute preincubation step of enzyme, NADPH, and inhibitor. The reaction is started by the addition of substrate, DTNB or GSSG, and a second aliquot of NADPH.

While compounds **6-8** caused maximal inhibition of TGR within 15 min (referred to here as fast inhibitors; Table 1, Fig. 3c), compounds **1-5** were found to be slow inhibitors as they displayed little to no inhibition after 15 min, but time-dependent inhibition of TGR over 25 h (Fig. 3d and Table 1).

Steady state studies varying NADPH and inhibitor concentrations indicate that treatment with the fast inhibitors **7** and **8** or the slow inhibitor **3**, (the only slow inhibitor reaching equilibrium in 6 h), resulted in decreases in both $K_m$ and $V_{max}$ values consistent with uncompetitive inhibition of TGR, while treatment with **9**, a close analog of compound **8** synthesized ad hoc to facilitate structural studies (see below), changed only the $V_{max}$ indicating noncompetitive inhibition (Supplementary Tables 15 and 16). Determination of steady state parameters for all the compounds was not possible due to low solubility at high micromolar concentrations. The change from uncompetitive to non-competitive mechanism induced by chemical modifications of the inhibitors is not rare in drug design studies[48–50]. In general, both uncompetitive and noncompetitive inhibitors exert their action through the binding of the ES complex and/or downstream catalytic species, with the difference that noncompetitive ones bind also to free enzyme. In TGR, and more generally in TrxRs, electron transfer from NADPH to the enzyme is fast and practically irreversible and so an actual ES complex (NADPH-TGR) is not significantly populated during the catalytic cycle[17]. We refer to the species formed after NADPH binding (the ES downstream species) as NADP$^+$-TGR(H) reduced complexes, indicating that electrons are inside the polypeptide chain of the enzyme giving rise to the EH$_2$ and the EH$_4$ species (Fig. 1a). To obtain orthogonal proof that our inhibitors bind to the NADP$^+$-TGR(H) forms, we evaluated their effect on thermal stability of TGR using the thermal shift assay (TSA or differential scanning fluorometry). A modification of the assay developed for flavoproteins has been reported and was utilized here[51]. Oxidized TGR had a $T_m$ of 73.7 °C, whereas TGR reduced with NADPH displayed a $T_m$ of 66.5 °C (Fig. 3e), indicating destabilization of the polypeptide chain upon enzyme reduction. Neither fast **6-9** nor slow TGR inhibitors **1, 3, 5** affected the $T_m$ of oxidized TGR. On the other hand, in the presence of NADPH there was an increase in the $T_m$ of 3.5–7.2 °C providing further evidence that the ES complex is the target of the inhibitors. The same time dependence on the shift in $T_m$ was seen for the compounds; fast inhibitors affected the $T_m$ in 15 min., while the slow inhibitors required >2 h to cause a shift in $T_m$. Structurally similar non-inhibitors **11** and **12** had no effect on the $T_m$ of either oxidized or reduced TGR. Next, we evaluated photocrosslinking between a photoreactive probe (PRP) (a slow inhibitor of TGR, Fig. 1d, Table 1) and recombinant TGR in the presence or absence of NADPH (Supplementary Fig. 24) using procedures we published previously[52]. In the presence of NADPH, labeling of TGR was more effective at both 5 and 50 µM of PRP than that without NADPH, indicating a higher yield of TGR-PRP adduct when NADPH was present. A control protein, *S. mansoni* histone deacetylase 8, was not labeled with or without NAPDH. Overall, the experiments clearly demonstrate preferable binding of our inhibitors to the NADP$^+$-TGR(H) complexes.

### Inhibitors do not convert TGR into an NADPH oxidase
Several covalent inhibitors of TGR are electrophilic compounds reacting with the Sec residue at the C-terminus, which induce a transition in the enzyme from an antioxidant to a pro-oxidant with increased NADPH consumption[20,45]. As expected, Stattic and TRi-1 converted TGR to a pro-oxidant enzyme, whereas our non-covalent inhibitors did not increase NADPH consumption or superoxide production (Fig. 3f, g), indicating again the lack of involvement of the Sec-containing C-terminus in the mechanism of action of these TGR inhibitors.

### TGR inhibitors bind to the doorstop pocket
To gain additional insights into the mechanism of inhibition, we conducted a series of structural studies. Our attempts to obtain X-ray co-

crystal structures of TGR and inhibitors resulted in structures with no detectable density of the inhibitors, possibly owing to the limited solubility of the compounds in the crystallization conditions. To increase aqueous solubility of the compounds to facilitate structural studies, the cycloheptyl substituent in TGR uncompetitive inhibitor **8** was replaced with a sugar moiety, resulting in compound **9**, a non-competitive inhibitor and thus capable of binding the enzyme in absence of NADPH. However, co-crystallization trials failed again. We, therefore, used alternative approaches and have determined the structure of a high molecular weight TrxR subfamily member using cryo-EM, demonstrating the feasibility of this methodology to study this protein subfamily of importance in several human diseases (PDB ID 8A1R, EMD-15084) (Fig. 4, Supplementary Table 17, Supplementary Fig. 25). First, the TGR-**9** complex was subjected to negative staining TEM to assess quality of TGR particles and then, upon sample vitrification, to a cryo-EM operating at 200 kV (Supplementary Fig. 25). After structural refinement of the X-ray structure of TGR into the cryo-EM map obtained at 3.6 Å resolution (as estimated by the gold-standard Fourier shell correlation at 0.143 Supplementary Fig. 25 and Fig. 4a), additional electron density ascribable to the compound is present in the doorstop pocket (Fig. 4b). Upon placement of the compound into the cryo-EM map and structural refinement we find that **9** adopts two conformations in both subunits, spanning the three subpockets A, B and C (Fig. 4b, c). Conformational changes induced by compound binding are not detected at this resolution. The correlation coefficient (CC) of **9** in the two different conformations is in the 0.67–0.71 range, close to the CCs of the FAD cofactor in each subunit (CC = 0.73–0.74). The two conformations differ in the position of the polar sugar moiety and in the orientation of the indole that is 180° rotated with respect to each other (Fig. 4b, c). In both conformations, the pinane ring of the compound interacts with F324, V469, T471, A481 and L484 in subpocket C (Supplementary Fig. 26), the indole interacts with F324, G325, P439 and L441 present in subpocket B (Fig. 1), while the sugar moiety, through its hydroxyl groups, is close to the main chain carbonyls of G323 and R322 in one conformation and with the analogous groups of G437 and Q440 occupying the hydrophilic portion of subpocket A in the other conformation. In agreement with the noncompetitive behavior of **9** and as determined by structural superposition of the cryo-EM structure and the X-ray NADPH-TGR- complex[17], inhibitor **9** does not sterically interfere with NADPH binding. Instead, it contacts F324, a conserved residue in GRs and high molecular weight TrxRs involved in the recognition of the oxidized nicotinamide moiety of NADP$^+$[53], suggesting that **9** may interfere with NADP$^+$ release and/or with the structural changes associated with it, slowing down the oscillations between EH$_2$ and EH$_4$ during enzyme turn-over (Fig. 5).

### TGR inhibitors can achieve selectivity over mammalian enzymes
Compounds **1-8** affected viability of Vero cells (EC$_{50}$) mostly at concentrations >50 µM (Table 2). An analysis of the raw data shows that for **1, 4-7,** and **9**, EC$_{50}$ against Vero cells is likely to be >200 µM. A more accurate assessment of these values was not possible due to low solubility of these compounds at high micromolar concentrations. To gain insights into the mechanism of VERO cell toxicity at high compound concentrations, the compounds were tested for inhibition of human cytoplasmic TrxR1 and GR. While none of the compounds show any appreciable inhibition of GR, TrxR1 IC$_{50}$ values varied from 4.9 to >50 µM (Table 1). Compound **6** is at least 25 times more potent against TGR than human TrxR1 indicating that selective enzyme inhibition is possible.

### TGR inhibitors are potent schistosomicidal agents ex vivo
Compounds in Table 1 were tested for schistosomicidal activity against *S. mansoni* and *S. japonicum* adult worms and *S. mansoni* NTS and 21-day juvenile worms ex vivo (Table 2). Schistosomicidal LD$_{50}$s against both species of adult worms were between 9.36 and 32.6 µM, juvenile

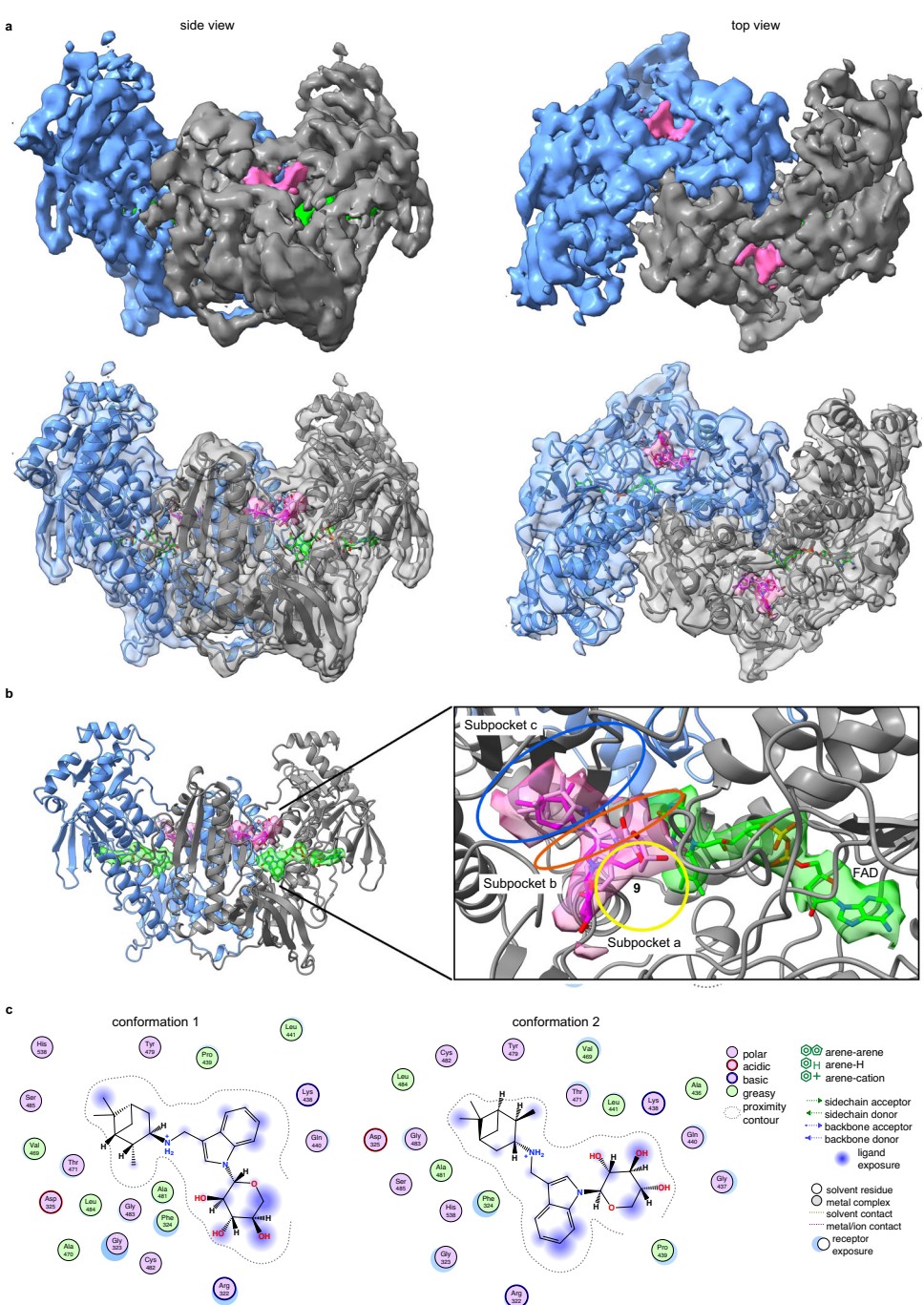

**Fig. 4 | Cryo-EM structure of the TGR-9 complex. a** Refined cryo-EM map of the TGR-**9** complex at 3.6 Å and its superposition with the resulting PDB model. The additional density ascribed to compound **9** and to the FAD are shown as pink and green continued surfaces, respectively. **b** Magnification of the doorstop pocket with **9** in two orientations (conformation 1 in magenta sticks; conformation 2 in pink sticks); the subpockets A-C are indicated as well as FAD shown as green sticks. **c** Interactions of the two conformers of compound **9** with surrounding amino acids.

worms 7.2 to >30 μM, and 0.6 μM to 17.5 μM against NTS, with most NTS LD$_{50}$ values below 6 μM. No differences in response were observed between male and female adult worms for any of the compounds tested. TGR inhibitors **1-6** were found to have higher potency against *S. mansoni* adult and NTS worm stages than PZQ and MZM, drugs with schistosomicidal activity. The PZQ LD$_{50}$ for NTS was determined after just 24 h exposure using both the Cell TiterGlo (27.1 ± 2.27 μM, Table 2) and the phenotypic analysis (42.9 ± 0.83 μM, Supplementary Movie 2), which is also used by others[54]. If incubation with compounds is prolonged, superior efficacies can be reached: after 72 h exposure of NTS to compounds **1** and **6** had LD$_{50}$s of 2.2 ± 0.19 and 7.8 ± 2.36 μM

respectively, whereas **2** had 0.18 ± 0.007 μM. In the assay with juvenile *S. mansoni* worms, TGR inhibitors **1-5** displayed potency comparable to that against adult worms and superior to PZQ and MZM, whereas LD$_{50}$ for **6-8**, PZQ and MZM were all >30 μM. Potency of inhibitors **1-4, 7**, and **8** against *S. mansoni* and *S. japonicum* adult worms was generally comparable.

**Efflux may affect efficacy of TGR inhibitors ex vivo**
Compounds **7-9** were found to have significant NTS killing activity, but the decrease in GSH/GSSG ratio was attenuated and less adult schistosomicidal activity was observed. Helminths are known to possess

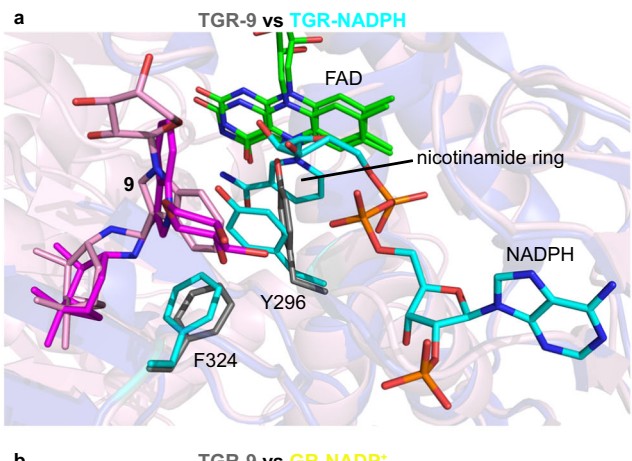

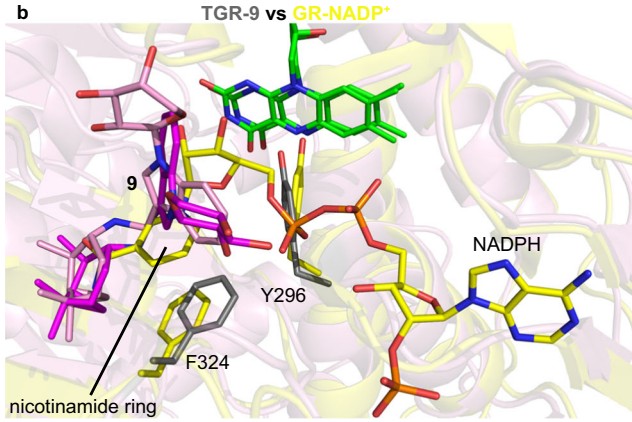

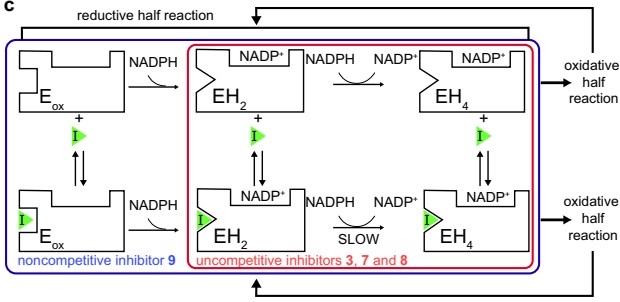

**Fig. 5 | Proposed mechanism of inhibition for the compounds described in this study. a** structural superposition between TGR in complex with **9** (in grey, PDB ID 8A1R) and TGR in complex with NADPH (in cyan, PDB ID 2 × 99). **b** structural superposition between TGR in complex with **9** and human GR in complex with NADP$^+$ (in yellow, PDB ID 3DK4). In both panels the FAD cofactors belonging to the different enzymes are in green sticks, while the two conformers of **9** are in pink and in magenta sticks. **c** hypothetical mechanism of inhibition for the noncompetitive (**9**) and uncompetitive (**3**, **7**, and **8**) inhibitors, considering the possibility that the destabilized NADP$^+$-TGR(H) reduced forms of the enzyme may specifically accommodate uncompetitive inhibitors due to the presence of conformational states more suitable for compound binding. EH$_2$ and EH$_4$ represent the reduced species with 2 and 4 electrons, respectively, populated during the enzymatic reduced half-reaction; EH$_4$ is the species competent for substrate reduction in the oxidative half-reaction. During catalysis, TGR oscillates between the 2-electron reduced state and the 4-electron state. The mechanism has been depicted considering the catalytic mechanism of TGR[17,30].

ABC/MDR transporters, P-glycoprotein, and other efflux transporters that mediate the transport of molecules and antimicrobials across the membrane modulating drug susceptibility[55]. To enhance the level of uptake and retention, we coupled treatments of **7-9** with channel blockers in ex vivo worms. Treatment with combinations of channel blockers, tariquidar (10 μM) and Ko143 (10 μM), resulted in increased adult worm killing compared to treatments without channel blockers (Table 3). Likewise, significant decreases in GSH/GSSG ratio in combination treatments compared to TGR inhibitors alone were also seen. Treatment with the channel blockers alone or in combination with negative control **11** resulted in insignificant decrease of the GSH/GSSG ratio and no effect on adult worm viability.

**TGR inhibitors have efficacy as schistosomicidal agents in vivo**
Encouraged by the activity of our non-covalent TGR inhibitors against ex vivo worms, we evaluated the schistosomicidal activity of these compounds in mice infected with *S. mansoni* (Fig. 6). A single treatment of mice 42 days post infection (d.p.i.) with **1** (100 mg/kg i.p.) resulted in a 44% decrease in worm and a 40% reduction in egg burdens, respectively. Administration, at 42 d.p.i., of two doses of **1** (100 mg/kg i.p. bid) resulted in 54% and 48% decrease in worm and egg burdens, respectively. A single dose of **2** (100 mg/kg i.p.) at 42 d.p.i. resulted in a 85% decrease in worm burden and 73% decrease in egg burden. Decreased efficacy was observed when this dose was spread over two administrations (50 mg/kg i.p. at 42 d.p.i. and 50 mg/kg i.p. at 45 d.p.i.), resulting in a 43% decrease in worm burden and a 69% decrease in egg production. A single dose of **6** (100 mg/kg i.p.) at 42 d.p.i. resulted in a 34% decrease in worm and 3% decrease in egg burdens, respectively, while two doses (100 mg/kg i.p. at 42 d.p.i. and 100 mg/kg i.p. at 45 d.p.i.), did not result in increased efficacy (38% decrease in worm and 18% reduction in egg burdens, respectively). A single administration of **8** at 42 d.p.i (200 mg/kg i.p.) resulted in a 19% and 7% decrease in worm and egg burdens, respectively.

The three TGR inhibitors with significant in vivo efficacy (**1**, **2**, and **6**) against adult worms were tested for efficacy against juvenile worms 21 d.p.i. (Fig. 6). As shown in previous studies, PZQ has very little activity in this context[13]. Treatment with either compound **1** (2 ×100 mg/kg) or **2** (1 ×100 mg/kg) resulted in significant decreases in both worm and egg burdens of 61% and 57% for **2** and 28% and 31% for **1**, respectively. Compound **6** exhibited only marginal efficacy of 16% and 7% decrease in worm and egg burdens, respectively. Thus, the greater efficacy of inhibitor **2** observed against adult worms, translated to the greater efficacy in mice against juvenile worms.

**Pharmacokinetic (PK) studies**
Administration of **1** (100 mg/kg i.p.) achieved a C$_{max}$ of 4.1 μM at 30 min and maintained plasma concentrations >2 μM at 2 h post administration: 4.1, 3.3, and 2.4 μM at 30, 60, and 120 min (Supplementary Table 18). A similar PK profile was observed for inhibitor **2** (100 mg/kg i.p.), with 4.8, 3.9, and 3.0 μM concentrations measured at 30, 60, and 120 min, respectively. Plasma exposure of mice to both **1** and **2** is compatible with observed efficacy and no overt toxicity was observed. Based on the time points available, the half-lives of **1** and **2** can be estimated at 90 min. The TGR inhibitor demonstrating greatest efficacy in treating infected mice, **2**, was also studied after oral administration: **2** (200 mg/kg p.o.) gave plasma concentrations of 0.71, 0.77, and 0.40 μM at 30, 60, and 120 min, respectively (Supplementary Table 19).

**Non-covalent inhibitors of TrxR class of redox proteins may have broad application**
One of our recently synthesized TGR inhibitors compound **10** with IC$_{50}$ = 18.6 μM for TGR is even more potent against TrxR from *Brugia malayi* (BmTrxR; IC$_{50}$ = 2.5 μM) and is a weak inhibitor of TrxR from *Plasmodium falciparum* (PfTrxR; IC$_{50}$ = 32.5 μM) (Table 1 and Supplementary Table 20). Both BmTrxR and PfTrxR are validated drug targets against lymphatic filariasis and malaria, respectively[56,57]. These results and the differences in the composition of the doorstop pockets (Supplementary Fig. 27) suggest that selectivity for individual TrxRs can be attainable, which may result in lower toxicity.

**Table 2 | Ex vivo characterization of 1–10, controls 11, 12, auranofin (AF), praziquantel (PZQ), and meclonazepam (MZM)**

| | ID | LD$_{50}$ (µM) ex vivo | | | | |
|---|---|---|---|---|---|---|
| | | *S. mansoni* adult | *S. mansoni* NTS | *S. mansoni* 21 days | *S. japonicum* adult | VERO |
| **Slow** | **1** | 12.3 ± 1.30 | 5.9 ± 0.25 | 26.3 ± 0.48 | 19.6 ± 1.91 | >75.5[a] |
| | **2** | 12.5 ± 0.40 | 0.6 ± 0.21 | 7.2 ± 0.30 | 11.1 ± 1.35 | 54.7 ± 6.65 |
| | **3** | 12.1 ± 2.44 | 4.3 ± 0.20 | 10.9 ± 0.39 | 13.9 ± 1.45 | 36.1 ± 3.0 |
| | **4** | 12.6 ± 1.86 | 0.85 ± 0.11 | 14.2 ± 2.13 | 27.1 ± 1.75 | >50[a] |
| | **5** | 9.36 ± 0.47 | 5.8 ± 0.15 | 16.9 ± 2.80 | n. d[b] | >50[a] |
| **Fast** | **6** | 15.7 ± 0.64 | 17.5 ± 3.7 | >30[a] | n. d.[b] | >100[a] |
| | **7** | 32.6 ± 0.26 | 4.7 ± 0.31 | >30[a] | 26.8 ± 1.62 | >50[a] |
| | **8** | 22.3 ± 0.2 | 9.4 ± 0.74 | >30[a] | 25.8 ± 1.87 | 32.7 ± 2.99 |
| | **9** | >100[a] | 11.14 ± 1.04 | n. d.[b] | n. d.[b] | >50[a] |
| | **10** | n. d.[b] | n. d.[b] | n. d.[b] | n. d.[b] | n. d.[b] |
| **Control** | **11** | >50[a] | >50[a] | n. d.[b] | n. d.[b] | n. d.[b] |
| | **12** | >50[a] | >50[a] | n. d.[b] | n. d.[b] | n. d.[b] |
| | AF | 1.1 ± 0.04 | 0.31 ± 0.02 | 0.62 ± 0.01 | n. d.[b] | 0.59 ± 0.11 |
| | PZQ | 32.8 ± 4.07 | 27.1 ± 2.27 | >50[a] | n. d.[b] | >50[a] |
| | MZM | 26.9 ± 0.97 | 13.7 ± 0.87 | >50[a] | n. d.[b] | >50[a] |

Schistosomicidal activity (LD$_{50}$, µM) of **1–12**, AF, PZQ and MZM determined against *S. mansoni* newly transformed schistosomula (NTS) (*n* = 200, at 24 h exposure), adult worms (*n* = 10, at 24 h exposure) and juvenile worms (21 days) (*n* = 10, at 48 h exposure) and *S. japonicum* adult worms (*n* = 10, at 48 h exposure). Cytotoxic activity of **1–9**, AF, PZQ and MZM against Vero cells (African Green Monkey Kidney cells, ATCC CCL-81) (*n* = 10$^4$ cells/well) after 24-hour exposure. Data are represented by *n* = 3 independent experiments as mean ± SD. Source data are provided as a Source Data file.
[a]greater than highest concentration tested (µM).
[b]*n.d.* not determined.

**Table 3 | Biological activity of compounds 7–9 and 11 alone and with channel blockers tariquidar (T) and Ko143 (K) both at 10 µM**

| Treatment | Decrease (%) in GSH/GSSG | Adult worm LD$_{50}$ (µM) |
|---|---|---|
| **7** | 36.9 | 32.6 ± 0.26 |
| **7** + TK | 69.5 | 20.6 ± 0.31 |
| **8** | 11.7 | 22.3 ± 0.20 |
| **8** + TK | 66.9 | 14.7 ± 0.97 |
| **9** | 9 | >100 |
| **9** + TK | 17.5 | 47.6 ± 8.5 |
| **11** | 0.4 | No effect |
| **11** + TK | 5.8 | No effect |
| TK | 5.0 | No effect |

Adult worm data presented by three independent experiments as mean ± SD of biological replicates.
Source data are provided as a Source Data file.

## Discussion

We have identified compounds (**1-10**) that act as non-covalent inhibitors of TGR with druglike properties as demonstrated by efficacy in mice infected with *S. mansoni*. Inhibition of TGR was shown in biochemical assays with recombinant protein, and in ex vivo worms by measurement of both TGR-generated fluorescent products from TRFS-Green and the decrease in the GSH/GSSG ratio. Herein, we demonstrate that single particle cryo-EM can be applied to a member of the pyridine nucleotide-disulfide oxidoreductase protein family, which includes crucial drug targets for several human diseases. The cryo-EM data show inhibitor **9** bound in the doorstop pocket, indicating that the design strategy, based on the initial fragments found in X-ray co-crystal structures, is successful. We propose that this class of inhibitors trap the NADP+-TGR(H) species, preventing NADP+ release. The evidence is provided by: (i) the inhibitors are found in the secondary site known to interact with outgoing NADP+ in related pyridine nucleotide–disulfide oxidoreductases;[53] (ii) inhibition is reversible; and (iii) inhibition is exerted by binding to the NADP+-TGR(H) reduced species. One advantage of noncompetitive and uncompetitive inhibitors over competitive inhibitors in disruption of metabolic pathways, is the lack of competition with endogenous substrates that may be present at high cellular concentrations as a result of enzyme inhibition, making uncompetitive inhibitors, in particular, ideal for drug development[58].

All lines of evidence indicate that fast and slow TGR inhibitors bind different conformational states present in the TGR enzymatic cycle induced by both NADPH-dependent reduction and the concomitant destabilization of the polypeptide chain, as shown by TSA. We hypothesize that (i) the slow inhibition observed in the biochemical assay with the recombinant protein with some of the compounds is due to their preferential binding to a slowly populated conformer of the NADP+-TGR(H) complex and (ii) this conformer is already populated in the worm cell interior accounting for the more rapid inhibition of TGR observed in ex vivo worms with the slow inhibitors. To the best of our knowledge, the assays where DTNB, GSSG, Trx, and NADPH (at saturating concentrations) are utilized as substrates are the only assays used to measure activity of TGR and related enzymes, including human TrxR and GR[59]. Considering that very little is known about the local and temporal concentrations of NADPH and TGR and the multiple redox-associated conformational states, it is difficult to assess how accurately the biochemical assay with recombinant TGR models the inhibition of TGR in worms ex vivo or in vivo. A comprehensive characterization of enzyme inhibition by slow inhibitors, which are well documented in the literature[58], is not straightforward. Hence, it certainly appears that measurements in worms ex vivo of GSH/GSSG ratio and fluorescent product from TRFS-Green are more reliable in assessing endogenous inhibition of TGR. The observed inhibition of GSSG reduction likely leads to redox stress resulting in the potent schistosomicidal activity observed. Despite similarity, TrxR and TGR proteins appear to be sufficiently different, as evidenced by the differential activity of our inhibitors against TGR and human, *B. malayi*, and *P. falciparum* TrxRs, and inhibitors selective for individual enzymes could be developed. Human TrxR1 (the cytosolic isoform) displays 74% sequence identity in the doorstop pocket residues with respect to TGR. Remarkably, the charge distribution and shape of TrxR1 in this site is different with respect to TGR[33] due to the presence of charged and bulky residues, i.e., E337, D338, E341, E368 and K389 in human TrxR1 in place of A436,

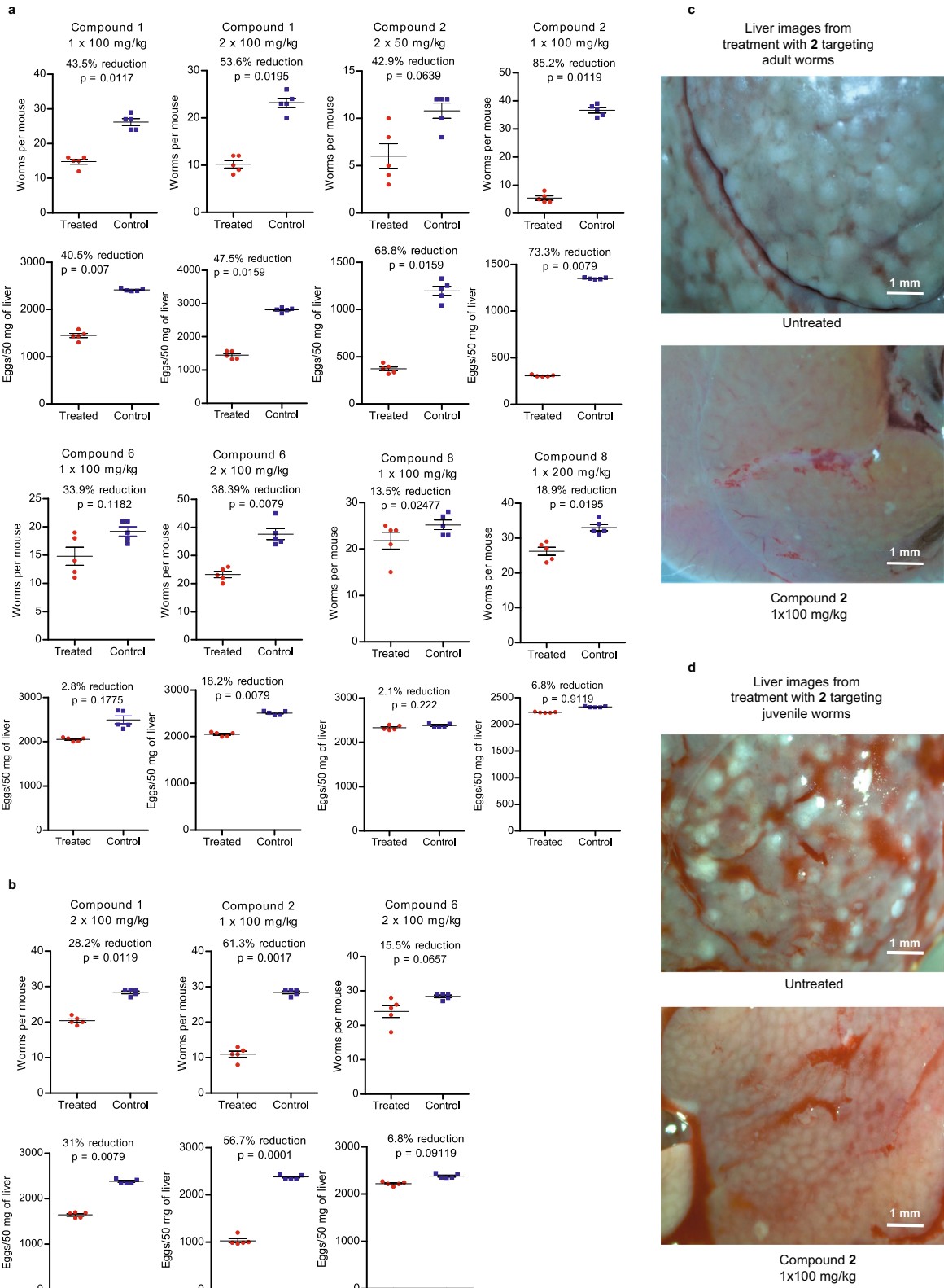

**Fig. 6 | Schistosomicidal efficacy in mice. a** Adult worm and liver egg burdens after compound treatments targeting adult worms 42 days after infection. **b** Adult worm and liver egg burdens after compound treatments targeting juvenile worms 21 days after infection. **c** Images of livers from a mouse treated with compound **2** at 100 mg/kg 42 days after infection and from an untreated mouse, showing reduction in the number of granulomas. **d** Images of livers from a mouse treated with compound **2** at 100 mg/kg 21 days after infection and from an untreated mouse, showing reduction in the number of granulomas. The number of mice in each treatment, $n = 5$. A two-tailed student t-test was used to determine significance, with the cutoff set to $p < 0.05$ in each comparison between mice treated with inhibitors and the control. Source data are provided as a Source Data file.

G437, Q440, S467 and D488 in TGR. These structural differences suggest that selective inhibition of TGR over human TrxR may be attainable. Indeed, compound **6** has $IC_{50} = 2.5\,\mu M$ against TGR and against human TrxR1 it is $> 66.7\,\mu M$, at least 25 times more potent against TGR than human TrxR1. Human Trx and GSH systems have compensatory activities so that inhibition of one arm can be supplemented by the other[60], suggesting that the inhibition of hTrxR by some of the compounds presented here will not be detrimental for humans as it is for schistosomes. Better understanding of the mechanism and TGR/TrxR reduced species involved in the binding of inhibitors depending on microenvironment may offer an additional avenue to control inhibitor selectivity in organism, tissue, and cell type specific manner.

The TGR inhibitors, reported herein, are schistosomicidal both ex vivo and in vivo. These inhibitors outperform PZQ, the drug of choice to fight schistosomiasis. Disease eradication using PZQ monotherapy is severely limited by the low activity of this drug against juvenile worms. Against ex vivo *S. mansoni* juvenile worms, the $LD_{50}$ of PZQ ($413\,\mu M$) is inferior to that of inhibitors **1-5** ($7.2 < LD_{50} \le 26\,\mu M$)[61]. Under the assay conditions used herein (ex vivo worm viability assessed after 24 h exposure), all compounds had equal or superior potency to PZQ and all, except **9**, were more active than MZM against *S. mansoni* adult worms and NTS (Table 2). The reported $LD_{50}$ for NTS after exposure to PZQ for 3 days is $1-2\,\mu M$[62]. After 3-days exposure of NTS, the $LD_{50}$ of compound **1** was similar to that of PZQ, whereas **2** was 5-fold more potent than PZQ. In other studies, the $LD_{50}$ for PZQ against adult worms has been reported as $1.5\,\mu M$ and $5.1\,\mu M$ for male and female worms, respectively;[61] however, these $LD_{50}$s were determined after overnight exposure to PZQ followed by 8 days culture. Our comparison of PZQ activity against NTS in the phenotypic and Cell Titer Glo assays found similar $LD_{50}$s immediately after 24 h exposure. The compounds presented herein have similar activities against both *S. mansoni* and *S. japonicum* adult worms and show no differences between male and female parasites. The similar effect observed against *S. japonicum* can be rationalized by the 100% homology of the residues in the doorstop pocket of TGR from the two species (Supplementary Fig. 27). The complete reliance of all schistosome species[15] on TGR for regulation of the redox defense network, and our previous results with several covalent TGR inhibitors[19,20], are compatible with species and sex concordance. Although some inhibitors (**7-9**), on the basis of adult schistosomicidal activity in combination with efflux transport blockers, appear to be substrates for efflux pumps, this does not cause species nor sex differences.

When targeting adult worms in mice, administration of inhibitors **1**, **2**, and **6**, resulted in significant reductions in both worm and egg burdens. Treatment with a single dose of compound **2** (100 mg/kg) resulted in an 85% reduction in adult worm burdens. The WHO criterion for lead progression is ≥80% reduction in worm burden after five doses (100 mg/kg qd) over multiple days[63]. This criterion is somewhat at odds with the current SOP for MDA, which does not entertain multiple doses over multiple days[27]. Compound **2** is therefore a viable development lead for treatment of schistosome infection. Earlier studies found that the $ED_{50}$ for PZQ against adult worms in mice was 80 mg/kg;[61] the activity for **2** reported here of 85% reduction at 100 mg/kg indicates similar activity. Targeting juvenile worms at 21 d.p.i. with a single injection of **2** (100 mg/kg) resulted in >60% reduction in worm burden, a significantly higher efficacy than observed for treatment with PZQ in previous studies[13], which had 0% reduction at 21 and 28 d.p.i. (500 mg/kg) and reductions of 50 % and 17% at 21 and 28 d.p.i. (1,000 mg/kg), respectively. Titrating PZQ doses 28 d.p.i. resulted in $ED_{50} = 2,456$ mg/kg[61]. Based on the single dose tested for **2**, the estimated $ED_{50}$ is below 100 mg/kg. Comparing the schistosomicidal activity against juvenile worms with that of PZQ reported in previous in vivo studies[61], the inhibitors reported herein show superior schistosomicidal activity to PZQ at the worm developmental stage in mice. Overall, treatment targeting juvenile worms resulted in smaller decreases in worm and egg burdens, which could be associated with higher ABC and MDR transporter activity in juvenile worms than adult worms[55].

The maximum plasma concentration ($C_{max}$) reached for a standard dose of PZQ (1,500 mg), in healthy volunteers, is 0.16 µg/ml or 513 nM[64]. The PZQ plasma exposure, measured by $C_{max}$, is substantially lower than the $LD_{50}$ against adult worms and NTS reported by others[61,62], and substantially below the $LD_{50}$ for NTS killing measured herein ($27-43\,\mu M$). The half-life of PZQ in healthy volunteers (1.6 h)[64] is also much shorter than the 3-8 days assays used by others to determine $LD_{50}$ for worms and NTS ex vivo. Discordance between ex vivo potency and in vivo PK/PD has many potential causes, one of which, in the case of schistosomicidal agents, is the involvement of the host immune response. The schistosomicidal activities of PZQ and of potassium antimonial tartrate, a drug used in the past for schistosomiasis and a TGR inhibitor, are reduced in immunosuppressed mice[65], indicating a crucial role of the host immune system in the mechanism of action of schistosomicidal agents in vivo. There is every reason to argue that the efficacy observed for compounds **1** and **2** in vivo incorporates a similar role for the host immune response as seen for other schistosomicidal agents. It is therefore unsurprising that the $LD_{50}$ values for compounds **1** and **2** determined ex vivo have a similar discordance to that seen for PZQ when considering in vivo PK/PD: much like PZQ these compounds are schistosomicidal in vivo at sub $LD_{50}$ concentrations. Although the maximum plasma concentration is lower when compound **2** is administered PO ($0.77\,\mu M$) instead of IP ($4.8\,\mu M$), it is still 4.3-fold higher than $LD_{50}$ for ex vivo NTS determined in the commonly used 72 h assay, also indicating that TGR inhibitor **2** is orally bioavailable and the initial prediction of oral bioavailability based on the Lipinski et al.[41] criteria is correct. The observation that higher plasma concentrations of **2** can be achieved after oral delivery than for PZQ, coupled with the greater ex vivo potency of compound **2**, strongly suggest that further optimization will yield an orally bioavailable schistosomicidal agent that will be transformative in the clinical setting. Inhibitors reported herein are effective against all stages of schistosome development, decrease egg production, improve liver pathology, and are orally bioavailable.

All parasitic flatworms have a redox system similar to that of *S. mansoni* with TGR serving an obligate role in the absence of TrxR and GR; therefore, our therapeutic approach can be extended to other human and veterinary flatworms[66]. Given the mechanism of enzyme inhibition, our non-covalent inhibitors can be targeted to TrxR and can be applicable to development of therapeutics for a broad range of diseases. All TrxR enzymes require an NADPH-dependent reduction step to exert their function in vivo and TrxR is a therapeutic target for cancer and infectious diseases[21,67]. Several currently used cancer drugs exert their anticancer activity in part through covalent inhibition of TrxR[21] and two covalent TrxR inhibitors are currently in clinical trials[68]. TrxRs from filarial nematodes and malaria parasites have been validated as druggable[56,57] and have structures available[69,70]. The druglike and orally bioavailable compounds identified herein are active in animal models of schistosomiasis with a broader range of developmental stages targeted than the drug of choice, PZQ. Their mechanism of action is different from that of PZQ and combinations of PZQ and TGR inhibitors represent a promising approach to develop combination therapies essential for schistosomiasis elimination.

## Methods

### Computer-aided molecular design, synthesis, and characterization

All the computer-aided molecular design studies were performed in SZMAP/GamePlan[37] and vBrood[38] modules in OpenEye software and Molecular Operating Environment, MOE[39]. The 3D structures of TGR in

biological assembly (i.e., dimers) PDB:6FP4 and PDB:6FTC were downloaded from PDB. The proteins were subjected to the Structure Preparation procedure followed by addition of hydrogen atoms using the Protonate 3D application[71] with all the default settings in MOE. The energy of the resulting structure was minimized utilizing AMBER14:-EHT forcefield in MOE[72,73] with all the settings set default until the RMS gradient reached 0.001 kcal/mol/ Å². The resulting ligand-proteins complexes were aligned to match the position of the co-crystallized small molecule fragments. At this step, the water molecules found in the X-ray structures were removed. The co-crystallized fragments found in sub-pockets A-C and the protein were used as input for the SZMAP, Gameplan, and vBrood applications and visualization in MOE. To facilitate the calculations and analysis, an artificial chimera molecule was built using the X-ray fragments found in subpockets A-C in PDB:6FP4 and PDB:6FTC and connected by a short $CH_2CH_2$ linker. The resulting complex was additionally minimized as described above. The default settings for the SZMAP and Gameplan applications were used in all the calculations. SZMAP was used for analysis of energetics of stabilizing and destabilizing effects of water. The *szmap* command with keyword *-stbl* resulted in complex, apo, and ligand grids. Additional grids were generated using *-results_set max* keyword. To break-out the region displaced by the ligand, the *grid_comp* command was run on the results of the previous step. WaterColor and Water-Orientation VIDA extensions were applied to the results of SZMAP calculations to visualize the regions that favor polar and non-polar substituents. The water probes position, orientation, and energetics were determined for the chimera-TGR complex. The −1.5 kcal/mol and 0.75 kcal/mol cutoffs were used for the lower and upper cutoffs in the "Exclude free energy range" settings, the water molecules were visualized by energy. The grid point pruning was set to "more". The positions of the probes and the free energy values were exported in a PDB file and visualized in Pymol[74]. The *S. mansoni* X-ray structures of TGR 2X99, 2X8G, 2X8H[75], 6FP4, 6FTC[33], 6ZST, 6ZP3, 6ZLP, 6FMU, 6ZLB, 7B02, 7NPX[34], 2V6O[75], 6RTJ, 6RTO, 6RTM[36], 3H4K[76] were downloaded from PDB, aligned in MOE, water molecules in the doorstop pocket were extracted and saved for visualization in Pymol. The same color scheme was used to visualize the water probes. Gameplan was used to generate a set of hypotheses for polar and non-polar ligand modifications. All the settings in Gameplan were set default. The fragment library for the vBrood application was prepared using the CHOMP application to fragment the ChEMBL database[77] version CHEMBL25[78]. In the CHOMP application, no additional parameters were used except for the minimal required to run it (i.e. -in and -out). The total number of fragments in the resulting database was 17,697,078. vBrood was run to replace the piperazine portion of the molecule using the default settings in vBrood.

The synthesis and characterization of compounds 1–12 and PRP are given in Supplementary Methods, Supplementary Figs. 1–20, and Supplementary Tables 1–14.

**Photolabeling of TGR with PRP**
TGR (1 μM) or SmHDAC8 (1 μM) was incubated at concentrations of either 5 or 50 μM PRP with or without 100 μM NADPH totaling eight samples. The photocrosslinking and CuAAC click biotinylation were performed as previously described by our group[52]. After resolving samples on SDS-PAGE and nitrocellulose membrane transfer, western blot normalization was performed with the Licor Revert™ 700 Total Protein Stain Kit as described in their protocol. Protein was finally imaged with the Licor IRDye® 800CW streptavidin on the Odyssey Sa imager.

**Enzyme inhibition, kinetics, ex vivo and in vivo activity**
**Enzyme preparation and activity determination.** Recombinant TGR, human TrxR1, *B. malayi* (Bm)TrxR and human GR proteins were expressed and purified as described[20,69,79]. The codon optimized

sequence for *P. falciparum* (PfTrxR, https://www.ncbi.nlm.nih.gov/protein/CAA60574.1/) with an N-terminal 6-His tag was synthesized and inserted into pET15b (GenScript) and expressed in BL21 (DE3) cells and purified as described for human GR[20]. TGR and TrxR enzyme inhibition assays were performed in triplicate at 25 °C as described[20] in 0.1 M potassium phosphate (pH 7.4), 10 mM EDTA, 100 μM NADPH and 0.01% Tween-20. TGR, human TrxR1 and BmTrxR (all at 4 nM) and PfTrxR (50 nM) were preincubated with the compounds for 15 min. The reaction was started with addition of an equal volume of DTNB (6 mM) and NADPH (100 μM) and the increase in $A_{412}$ during the first 3 min was recorded. To determine inhibition of GR, GR (120 pM) was added to an assay mixture (100 μM NADPH, 0.1 M potassium phosphate (pH 6.9), 200 mM KCl, and 1 mM EDTA). The reaction was pre-incubated for 15 min. Activity was initiated with the addition of 1 mM GSSG and 100 μM NADPH and initial rates of NADPH oxidation were monitored at 340 nm. The reactions were done in triplicate. The $IC_{50}$ was calculated in GraphPad Prism.

**Thermal shift assay (TSA).** Samples were prepared by using a final concentration of 0.25 mg/ml TGR diluted in TGR reaction buffer to the desired volume. To evaluate reduced TGR, 500 μM NADPH was added. Inhibitors were added at a concentration dependent on the compound's $IC_{50}$, from 125–500 μM. The mixtures containing inhibitors **1-5** were incubated for 6 h and for 30 min for treatments with **6-8** at room temperature. After the preincubation period, 20 μL samples were pipetted into a BioRad un-skirted PCR 96 well plate and sealed with MicroAmp™ Optical Adhesive film. The plate was then centrifuged for 5 min at 1000 × g. TSA was carried out using a BioRad CFX Connect qPCR instrument with a melt curve setting of 25 − 95 °C, in increments of 0.5 °C/10 seconds. The SYBR green channel was used to detect fluorescence as the wavelength of flavin fluorescence overlaps with that of SYBR green. BioRad CFX Maestro 5.2 was used to analyze TSA data.

**NADPH dependence of inhibition.** TGR was incubated at room temperature with inhibitor in the presence or absence of 100 μM NADPH for 15 min. DTNB (3 mM) and NADPH (100 μM) were added, and the reaction was monitored at $A_{412}$ for 5 min to determine reaction rate. The assay was done in triplicate.

**Time dependence of inhibition.** TGR was incubated at room temperature (up to 6 h) and at 4 °C for 6 to 24 h with 50 μM inhibitor and 100 μM NADPH for the indicated times. Then DTNB (3 mM) and NADPH (100 μM) were added, and the reaction was monitored at $A_{412}$ for 5 min. The assay was done in triplicate.

**Reversibility by jump dilution.** A reaction of 370 nM TGR, 100 μM NADPH, and 250 μM inhibitor was incubated for 15 min at room temperature. A 100x dilution of the reaction was made and the activity was determined immediately and after 60 min. The inhibition of 3.7 nM TGR with 250 μM inhibitor, 100 μM NADPH, and a 15 min pre-incubation at RT was measured to compare to the inhibition after the jump dilution.

**NADPH competition.** Master mixes with varying concentrations of NADPH were made, each with 2 nM TGR in reaction buffer. Inhibitor (2 μl) was added to a 96 well microplate in triplicate for each NADPH and inhibitor concentration, and 192 μl of the master buffer was added to each well. The reaction incubated for 15 min and 6 μl of 50 mM DTNB was added to each well. The kinetic rate was measured at $A_{412}$ for 5 min.

**Superoxide production.** The production of superoxide by TGR was determined by monitoring reaction of superoxide with pyrogallol red[80]. Briefly, a 1 ml reaction mix (500 nM TGR, 100 μM NADPH, and

20–50 μM inhibitor) was incubated at room temperature for 30 min for fast inhibitors, or 2 h for slow inhibitors. For irreversible inhibitor controls TRi-1 (synthesized in house as reported[45]), Stattic (Cayman Chemical), and AF (Cayman Chemical), the samples were desalted using a Zeba spin desalting column. 100 μL of sample was combined with pyrogallol buffer with and without superoxide dismutase (SOD) (50 μM pyrogallol red, 300 μM NADPH; ± 10 units SOD). The reaction was measured at $A_{340}$ and $A_{540}$ simultaneously for 2 h.

**NADPH consumption.** TRi-1, Stattic (both at 50 μM) and AF (20 μM) were incubated with 1 ml of 500 nM TGR and 100 μM NADPH in TGR reaction buffer for 30 min. The samples were desalted using a Zeba spin desalting column (Thermo Fisher Scientific) and 100 μl of the desalted sample was combined with 100 μl of 100 μM NADPH. NADPH consumption was monitored at 340 nm in triplicate. Compounds **1, 4, 6**, and **7** (50 μM) were tested in the same fashion without the desalting step and with a first incubation step of 30 min for **6** and **7** and 2 h for **1** and **4**.

### Evaluation of schistosomicidal activity
**Preparation of NTS.** *Oncomelania hupensis* subsp. *hupensis*, Chinese strain, infected with *S. japonicum*, Chinese strain, and *Biomphalaria glabrata*, strain NMRI, infected with *S. mansoni*, strain NMRI, were provided by the NIAID Schistosomiasis Resource Center for distribution through BEI Resources, NIAID, NIH. After infections were patent, snails were exposed to bright light for 1 hr to obtain cercariae. Cercariae were mechanically transformed to schistosomula[54]. Briefly, cercariae were placed on ice for 30 min and then centrifuged at 350 × g for 10 mins. The supernatant was decanted and 2 ml of serum-free M199 medium was added to cercarial pellets and vortexed for 1 min until cercarial tails were detached. NTS were purified by layering on 4 °C Percoll gradient suspension containing Eagle's minimum essential medium, penicillin-streptomycin (10,000 U per ml penicillin/10,000 U per ml streptomycin), and 1 M HEPES in 0.85% NaCl with cercariae suspension and centrifuged at 500 × g for 15 min. Cercarial pellets were resuspended and washed thrice in serum-free M199 medium and collected at 100 × g for 5 min. NTS (240) were transferred to U-bottom 96 well assay plates containing 200 μl of M199 medium supplemented with 5.5 mM D-glucose, penicillin-streptomycin and 5% heat inactivated fetal bovine serum and incubated at 37 °C in a 5% $CO_2$ incubator overnight.

**Preparation of juvenile and adult worms.** All animal studies at Rush University Medical Center were approved by the Institutional Animal Care and Use Committee of the Rush University Medical Center (Department of Health and Human Services animal welfare assurance number A-3120–01) with protocol ID: 20-069. Three-week old, female Swiss-Webster mice obtained from the Charles River were housed in the Comparative Research Center of Rush University Medical Center. Mice were infected by percutaneous tail exposure to about 200 *S. mansoni* or 50 *S. japonicum* cercaria for adult worms and about 1000 cercaria for juvenile *S. mansoni* worms through natural transdermal penetration of the cercariae for 1 h[81]. Mice were euthanized three-and seven-weeks post infection for juvenile and adult worms, respectively, using a lethal dose of 0.018 ml of Euthasol and 5.85 mg/ml heparin to prevent blood coagulation (injection volume of 400 μl). Perfusion was performed by flushing pre-warmed RPMI containing phenol red and L-glutamine through a 25- and 3/8-gauge needle placed into the aorta attached to Tygon tubing aided by the Masterflex L/S perfusion pump as described[81]. Juvenile and adult worms were carefully washed in phenol red free RPMI medium and subsequently incubated in phenol red free RPMI medium supplemented with 5.5 mM D-glucose, penicillin-streptomycin and 5% heat inactivated fetal bovine serum and at 37 °C in a 5% $CO_2$ incubator overnight.

**Schistosomicidal activity of compounds against NTS, juvenile and adult worms.** DMSO formulated compounds were diluted with phenol red free M199 medium or RPMI medium for NTS or juvenile and adult worms, respectively, at <1% DMSO final concentrations. NTS, juvenile and adult worms from overnight cultures were tested against compounds in triplicate. Controls were treated with DMSO alone or 5 μM AF as a positive control in appropriate medium[18]. Worm viability was assessed at 24 or 72 h by measuring ATP content of worms using Cell Titer Glo Assay (Promega) as described[82]. Schistosome viabilities in the presence of the compounds were assessed using this formula: % Viability = Averages of Test / Averages of DMSO Control x 100.

**Channel blockers enhanced schistosomicidal activity of compounds.** To assess the involvement of efflux pumps in the diminished schistosomicidal activity observed in selected compounds (**7**-**9**) against *S. mansoni* adult worms, we treated adult worms with these compounds in the presence of channel blockers Tariquidar (T, 10 μM, Cayman Chemical) and Ko143 (K, 10 μM, Cayman Chemical) as described above.

**Phenotypic assessment of PZQ activity on NTS.** About 200 NTS incubated at 37 °C in a 5% $CO_2$ incubator overnight were exposed to different concentrations of PZQ (1, 5, 10, 20, 30, 40 and 50 μM) and control without treatment for 24 h in triplicate. Worm images were acquired using Keyence BZ-X800 microscope and PZQ activity evaluated phenotypically[62]. NTS viability was assessed by scoring worms based on morphological changes and motility. Viability scores of 3 = motile, no changes to morphology and transparency, 2 = reduced motility and/or some damage to tegument as well as reduced transparency and increased granularity, 1 = severe reduction of motility and/or damage to tegument with high opacity and high granularity, and 0 = dead. The effect of PZQ on NTS was determined using the formula % Effect = 100 − (Average (Test) x 100 / Average (Control)) and $LD_{50}$ determined in GraphPad Prism.

**Cytotoxicity in mammalian cells.** Vero cells (African Green Monkey Kidney cells, ATCC CCL-81) were grown in Dulbecco's modified Eagle's medium (DMEM) containing glucose, L-glutamine and sodium pyruvate and supplemented with 10% fetal bovine serum and 1X penicillin−streptomycin (Sigma) at 37 °C in culture flasks (TPT-90025) until confluent growth was attained. Vero cells were detached from the flasks by treatment with trypsin (0.5 mg/ml)/EDTA (0.2 mg/ml) in PBS for 5 min at 37 °C. Detached cells were suspended in the modified DMEM medium and seeded at $10^4$ cells/well in 96-well microtiter plates (Costar, Corning) and incubated in the presence of 5% $CO_2$ at 37 °C for 24 h. Following overnight incubation, formulated DMSO-compound stock solutions were diluted with DMEM at <1% DMSO final concentrations. Vero cells were treated with different concentrations of compounds with DMSO as control and incubated in the presence of 5% $CO_2$ at 37 °C for 24 h. Vero cells treated similarly with different concentrations of PZQ, MZM and AF (Cayman Chemicals) were used as positive controls. Vero cell viability was assessed at 24 h by measuring ATP content using Cell Titer Glo Assay as described[83].

**Assessment of schistosomicidal activity in mice.** To assess the efficacy of the compounds in vivo against juvenile and adult worms, five female Swiss-Webster mice were randomly assigned for the control and experiment groups using a randomization tool embedded in GraphPad Prism. Mice were percutaneously exposed for 1 h to about 80 *S. mansoni* cercariae. Three- and six-weeks post infection, respectively, for juvenile and adult worms, mice were treated depending on their weight with formulations of the compounds, while the control mice received only the vehicle. Compounds were formulated with 10% DMSO and 10% Tween-80, and vortexed to obtain a uniform mixture. The mixture was sonicated twice for 5 min each time using a digital

ultrasonic cleaner heated to 50 °C. Sodium chloride 0.9% (80% normal saline) was added to the mixture, briefly vortexed and sonicated one more time using the same conditions as previous. The investigators were blinded by which group of mice received a treatment and the vehicle, as injection was done by technicians from the Comparative Research Center of Rush University Medical Center. Using a 26 G, 3/8 in intradermal bevel needle, each mouse was administered intraperitoneally with 100 μl of the formulated compound suspension at 50 mg/kg, 100 mg/kg and 200 mg/kg. Mice treated three weeks post infection for juvenile worms were euthanized three weeks post treatment and mice treated six weeks post infection were euthanized one week post treatment with 0.018 ml of Euthasol and 5.85 mg/ml heparin and perfused[81]. The mesenteric and hepatic portal veins of the mice were carefully scanned under the microscope to extract any remaining *S. mansoni* adult worms and worm burdens were determined. Egg burden was determined by weighing 50 mg of liver tissue from each of treated and control mice. The liver tissues were digested with 5% KOH at 37 °C overnight and washed twice with PBS. The number of eggs per 50 mg of liver tissue was determined using a Keyence BZ-X800 microscope using the egg autofluorescence[84].

**TGR inhibition in worms: TRFS-Green fluorescence quantification and GSH/GSSG determination.** TGR inhibition in NTS was assessed using a fluorescent probe TRFS-Green (Medchemexpress)[43]. NTS prepared as previously described and cultured in M199 supplemented with 5.5 mM D-glucose, penicillin-streptomycin and 5% heat inactivated fetal bovine serum were incubated at 37 °C in a clear bottom flat well plate in a 5% $CO_2$ incubator overnight. To inhibit TGR activity, NTS were treated with compounds (30 μM) or auranofin (5 μM) for 2 h. NTS were further treated with TRFS-Green (10 μM) for additional 4 h and rinsed with M199 medium to remove residual TRFS-Green. Fluorescence images were obtained using Keyence BZ-X800. The quantification of fluorescence intensity upon the uptake of TRFS-Green by NTS was performed by a fluorescent microplate imager (BioTek Cytation3) (excitation, 438 nm; emission, 538 nm) hourly for 5 h and after 24 h.

We assessed GSH/GSSG levels experimentally by treating *S. mansoni* adult worms with 50 μM of compounds, or respective controls 5 μM Auranofin, 50 μM PZQ or 50 μM MZM and 0.01% DMSO for 3 h. Worm homogenate was prepared by washing three times with PBS to remove residual compounds and ten volumes of ice cold 5% sulfosalicylic acid was added. Worms were manually homogenized on ice using VWR Pellet Mixer Adaptor. The worm suspension was centrifuged at 14000 × g, 4 °C for 10 min and the acid supernatants transferred. An equal volume of ice-cold neutralization buffer (500 mM HEPES, pH 8.0) was added to the acid supernatants.

GSH/GSSG assay was performed by diluting the acid supernatant (5-fold) with ice-cold dilution buffer (250 mM HEPES, pH 7.5). Using white opaque luminescence plate on ice, 25 μl of the diluted worm acid supernatant was added each well in triplicates. Total glutathione lysis reagent and GSSG lysis reagent, which contains alkylating agent N-ethylmaleimide (25 mM) were prepared following manufactures protocol and added to respective wells along with blank for background. Total glutathione lysis reagent or oxidized glutathione lysis reagent (25 μl) was added to respective wells containing worm acid supernatants and the plate was shaken for 5 min. Luciferin generation reagent (LGR) (50 μl) was added to all wells, briefly shaken and incubated at room temperature for 30 min. Luciferin detection reagent (LDR) (100 μl) was added to all wells, briefly shaken and incubated for 15 min at room temperature and the luminescence measured using microplate imager (BioTek Cytation3). GSH/GSSH ratio for DMSO control and compounds were calculated using (Net DMSO total glutathione RFU) − (Net DMSO GSSG RFU)/ (Net DMSO GSSG RFU)/2 and (Net inhibitor total glutathione RFU) − (Net inhibitor GSSG RFU)/ (Net inhibitor GSSG RFU)/2 respectively.

## Pharmacokinetics of 1 and 2 in vivo: intraperitoneal administration

**Chemicals and reagents.** HPLC-grade water was prepared by an in-house PURELAB Option filtration system (Elga lab water solution, UK). All reagents and solvents used were of HPLC grade. Methanol (VWR Chemicals), formic acid (Sigma-Aldrich, West Chester, PA, USA) & DMSO Sigma-Aldrich (West Chester, PA, USA), NADPH (Merk, USA).

**Animals.** Swiss Webster male mice were purchased from Charles River Laboratories (Wilmington, MA, US). At 6 weeks of age, mice were housed in plastic cages and received standard chow (AIN-76) and water ad libitum prior to experiment maintained on a 12 h/12 h light/dark cycle. All the mice were weighed and dosed intraperitoneally accordingly at 100 mg/kg body weight with freshly prepared **1** or **2**.

Blood was collected by submandibular puncture into 1 mL Eppendorf tubes, the plasma was separated and stored in −80 °C until used. A 25 μl of mouse plasma was transferred into 1.5 ml centrifuge tube which was spiked with 2.5 μl of internal standard (IS) working standard solution (100 μg/ml) to get final IS contraction of 250 ng/ml and the solution was vortexed. An aliquot of 75 μl of methanol/0.15 formic acid was added and gently vortexed for 2 min. The samples then centrifuged for 20 min at 4 °C and 15000 × g. A clear supernatant (50 μl) with added 50 μl mobile phase (reconstitution solution) from each extraction was then transferred into a 250 μl autosampler vial. A 3 μl aliquot of each sample was injected for LC-MSMS analysis.

Animal experiments were performed according to the policies and guidelines of the Institutional Animal Care and Use Committee (IACUC) of the University of Illinois at Chicago (Protocol 19-049).

**LC-MS/MS conditions.** The analyte molecules were eluted on Zorbax XDB-C18 column (3.5 μm, 2.1x20 mm) with the mobile phase composed of water/0.1% formic acid and methanol/0.1% formic acid in the ratio of 70:30 V/V with the flow rate of 0.2 ml/min. The total run time of 2 min with an injection volume of 3 μl at column temperature 40 °C were efficient to achieve accepted results.

A Shimadzu LC20AD, Ultra Performance Liquid Chromatography (UPLC) system (Shimadzu Corporation, Kyoto, Japan) equipped with Shimadzu 8040 triple quadrupole (QqQ) mass analyzer (Shimadzu Corporation, Kyoto, Japan) with an electrospray ionization source (ESI) operated in the positive charge mode for the quantification of **1** and **2**.

Instrument control and data acquisition was achieved via LabSolutioFns software (Shimadzu Corporation, Kyoto, Japan).

## Pharmacokinetics of 2 in vivo: oral gavage

**Chemicals and reagents.** HPLC-grade water was prepared by an in-house PURELAB Option filtration system (Elga lab water solution, UK). All reagents and solvents used were of HPLC grade. Methanol (VWR Chemicals), formic acid (Sigma-Aldrich, West Chester, PA, USA) & DMSO Sigma-Aldrich (West Chester, PA, USA), NADPH (Merk, USA).

**Animals.** Compound **2** was dissolved in DMSO. Tween 80 and saline were then added to the DMSO solution in that order. Final solution was 10% DMSO:10% Tween 80:80% saline.

Swiss Webster female mice were purchased from Charles River Laboratories (Wilmington, MA, USA). At 6 weeks of age, mice were housed in plastic cages and received standard chow (AIN-76) and water ad libitum prior to experiment maintained on a 12 h/12 h light/dark cycle. All the mice were weighed and gavaged orally at 200 mg/kg body weight with freshly prepared **2**. Blood was collected by submandibular puncture at 0, 0.25, 0.5, 1, and 2 h.

Animal experiments were performed according to the policies and guidelines of the Institutional Animal Care and Use Committee (IACUC) of the University of Illinois at Chicago (Protocol 19-049).

**Analytical standard preparation.** The multiplexed working standards were prepared by dilution from stock solutions in methanol at concentrations from 2.5 ng/ml – 2000 ng/ml. The working internal standards were also diluted from stock solutions in methanol with 0.1% formic acid. The working concentration of internal standard compound **1** was 750 ng/ml.

**Sample preparation.** Calibration curve for **2** was prepared by spiking 40 μl of blank mouse plasma with 10 μl of working standard solution to get final concentration ranging from 2.5 ng/ml – 2000 ng/ml. Samples were deproteinized with 150 μl of internal standard solution. This mixture was vortexed to mix properly and centrifuged at 13000 × g for 30 min. 160 μl of the supernatant was transferred to another set of tubes and dried using a nitrogen evaporator. Samples were reconstituted in 100 μl of methanol, vortexed and centrifuged at 13000 × g for 20 min. 30 μl of supernatant was transferred to autosampler vials.

**LC-MS/MS conditions.** Experiments were carried out on Thermo-Fisher Scientific Vanquish high-performance liquid chromatography connected to ThermoFisher Scientific TSQ Quantis – triple quadrupole mass spectrometer. The analyte molecules were separated on Phenomenex kinetex C18 column 100 Å (2.6 μm, 50 × 3 mm) with the mobile phase composed of Water/0.1% formic acid (A) and Acetonitrile (B) at the flow rate of 0.5 ml/min with the column temperature maintained at 40 °C and the sample of injection was 5 μl. The gradient program was set as follows: 0.1 min, 5% B; 0.1–0.5 min, 5–95% B; 0.5–2.5 min, 95% B; 2.5–3.5 min, 95–5%; 3.5–4 min and stop at 5 min.

Quantification was performed using electrospray in the positive mode with the spray voltage of 3500 V. Sheath gas (Arb) 20, Auxiliary gas (Arb) 8 and Sweep gas (Arb) 5.7. The ion transfer tube has a temperature of 325 °C and vaporizer temperature of 350 °C.

A Shimadzu LC20AD, Ultra Performance Liquid Chromatography (UPLC) system (Shimadzu Corporation, Kyoto, Japan) equipped with Shimadzu 8040 triple quadrupole (QqQ) mass analyzer (Shimadzu Corporation, Kyoto, Japan) with an electrospray ionization source (ESI) operated in the positive charge mode for the quantification of **2**.

Instrument control and data acquisition was achieved via LabSolutions software (Shimadzu Corporation, Kyoto, Japan).

**Evaluation of compound 2 stability under the biochemical assay conditions with recombinant TGR**

TGR assay solutions (1 ml each) were prepared as described in the main manuscript. Compound **2**, NADPH, and incubation time was varied as following: 1) TGR and **2**, no preincubation, 2) TGR, **2**, and NADPH, no preincubation, 3) TGR and **2**, 15 min preincubation, 4) TGR, **2**, and NADPH, 15 min preincubation, 5) TGR and NADPH followed by 15 min preincubation, **2** was added after 15 min preincubation, 6) TGR and **2** followed by 15 min preincubation, NADPH was added after 15 min preincubation. Each of the reaction mixtures were terminated by the addition of 250 μl of ethyl acetate, extracted with 1 ml of methyl tert-butyl ether. The organic layer was evaporated in vacuo, and the residue was redissolved in 0.5 mL of 50% MeOH before injection onto LC column. LC-MS analysis was done on a Waters SYNAPT quadrupole/time-of-flight mass spectrometer operated in positive ion electrospray mode. The column was Waters XBridge C8 column and gradient was from 20-90% MeCN/0.1% formic acid over 12 min. No additional ions except for those corresponding to **2** were detected.

**Assessment of reactivity of 2 and TRi-1 with N-Boc protected methyl ester of L-selenocysteine**

Dimethyl bis(*N-tert*-butoxycarbonyl)-L-selenocystine was synthesized as reported previously[85]. (*N-tert*-butoxycarbonyl)-L-selenocysteine

methyl ester was prepared in situ[86] in methanol or phosphate buffer. Compound **2** was added to the solution of (*N-tert*-butoxycarbonyl)-L-selenocysteine methyl ester in either methanol or phosphate buffer and incubated for 40 min, 24 h, and 120 h under N₂ gas balloon. Additional amounts of NaBH₄ were added at 12 h and 96 h to maintain the reducing conditions. Aliquots taken from methanol solution at each of the time points were concentrated in vacuo, mixed with water, and extracted with ethyl acetate. The organic layer was separated, concentrated in vacuo, re-dissolved in methanol and analyzed by LCMS. Aliquots taken from the phosphate buffer solution were extracted with ethyl acetate. The resulting organic layer was separated and concentrated in vacuo. The residue was re-dissolved in methanol and analyzed by LCMS. No additional peaks were detected either by UV or MS detection. No additional spots were detected by TLC as well.

Covalent TGR inhibitor TRi-1 was used as a positive control for the reaction with (*N-tert*-butoxycarbonyl)-L-selenocysteine methyl ester. The reactivity of TRi-1 was tested similarly as described above for compound **2**. An adduct between TRi-1 was detected by TLC and LCMS analysis (Supplementary Fig. 22).

## Cryo-EM methods

**Negative staining transmission electron microscopy.** The homogeneity of the protein before structural determination was assessed by negative staining electron microscopy. Around 4 μl of the mixture of 0.02 mg/ml TGR in 0.15% DMSO and 5 mM inhibitor was applied to home-made carbon film evaporated a mica film, which was floated off in about 200 μl of 2% sodium silicotungstate (SST) and recovered by a Cu grid. The stained sample was then air dried. The images were acquired on a Tecnai 12 (Thermo Fisher Scientific) LaB₆ electron microscope operating at 120 kV using a Gatan Orius 1000 CCD camera or on a Tecnai F20 (Thermo Fisher Scientific) FEG electron microscope operating at 200 kV using a Ceta CMOS camera (Thermo Fisher Scientific).

**Cryogenic electron microscopy.** The specimens for cryogenic electron microscopy (Cryo-EM) have been prepared onto 300 mesh Ultrafoil Au R1.2/1.3 grids (Quantifoil Micro Tools GmbH, Germany). First, 0.4 mg/ml TGR protein was mixed with 5 mM of inhibitor **9** in buffer solution containing 0.15% DMSO and incubated 30 min at room temperature (20 °C). Then, 3.5 μl of the sample was applied onto 45 s glow-discharged quantifoil grids and vitrified in liquid ethane using a Vitrobot Mark IV (ThermoFisher Scientific) at 100% humidity, 7 s blotting time and 10 s waiting time. A total number of 2635 raw movie stacks made of 50 frames each were collected with SerialEM[87] from a single individual session with a GLACIOS 200 kV FEG Cryo-TEM (ThermoFisher Scientific) using a K2 Summit detector (Gatan Inc., USA) at 36000× magnification and pixel size of 1.145 Å/pixel without pre-exposure and using a defocus of −2.6 to −1.8 μm and 50 e⁻/Å² total dose per stack (1 e⁻/Å² per frame). Single-particle structure determination has been carried out with Relion v3.1.2[88] after motion-correction using 5 × 5 patches[89] and CTF estimation[90]. About 10000 particles were picked using a Laplacian-of-Gaussian approach to build 2D templates from eleven selected motion-corrected micrographs for further template-based picking to reach more than 1.7 × 10⁶ particles. After extraction with 2-fold binning (216 to 108 pixels), 2D classification was used to eliminate wrongly picked particles. An initial 3D model was built in Relion with both C1 and C2 symmetry and used for subsequent 3D classifications and refinements. All classes obtained after 3D classification and refinement showed the best results in terms of overall resolution and appearance of electron density. The final rounds of 3D classification and 3D refinement were performed using a larger box size (300 pixels) without binning along with CTF refinements and particle polishing. The final 3D map has an average resolution of 3.6 Å at FSC = 0.143. The crystal structure of TGR (PDB ID: 2V6O)[32] was docked manually in the cryo-EM map using COOT[91,90]. Map local

anisotropic sharpening and real space refinement was carried out with Phenix[92], while manual model building was done with COOT. After several cycles of refinement and model building, the inhibitor was placed into the cryo-EM map.

### Reporting summary

Further information on research design is available in the Nature Portfolio Reporting Summary linked to this article.

## Data availability

The cryo-EM data generated in this study have been deposited in the PDB and in the EM data bank under accession codes 8A1R and EMD-15084, respectively. The X-ray-derived data of small molecular fragments in complex with TGR used in this study to design the compounds here described are available in the PDB under accession codes 6FTC, 6FMU, 6FMZ, 6FP4, 2X99, 2X8G, 2X8H, 6ZST, 6ZP3, 6ZLP, 6ZLB, 7B02, 7NPX, 2V6O, 6RTJ, 6RTO, 6RTM, and 3H4K. The X-ray-derived data of human GR in complex with NAPD+ used in this study to formulate the inhibition mechanism of the compounds depicted in Fig. 5 is available in the PDB under accession codes 3D4K. The ChEMBL25 data used in this study are available in the ChEMBL25 database here [https://doi.org/10.6019/CHEMBL.database.25]. Source data are provided with this paper.

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

## Acknowledgements

*Oncomelania hupensis* subsp. *hupensis*, Chinese strain, infected with *S. japonicum*, Chinese strain, and *Biomphalaria glabrata*, strain NMRI, infected with *S. mansoni*, strain NMRI, were provided by the NIAID Schistosomiasis Resource Center for distribution through BEI Resources, NIAID, NIH. We are grateful to Dr. Guy Schoehn (Univ. Grenoble Alpes, CEA, CNRS, Institut de Biologie Structurale, Grenoble, France), Prof. Beatrice Vallone (Sapienza University of Rome, Italy) and Dr. Linda C. Montemiglio (IBPM, National Research Council, Italy) for helpful discussions of the cryo-EM studies. We acknowledge the Elettra-Sincrotrone Trieste (Italy) for support in X-ray data collections and the European Synchrotron Radiation Facility for provision of microscope time on CM01. The study was funded in part by US NIH/NIAID R33AI127635 to F.A., P.A.P., G.R.T. and D.L.W. This work benefited from access to Research Resources Centre and UICentre at University of Illinois at Chicago and used the platforms of the Grenoble Instruct-ERIC center (ISBG; UAR 3518 CNRS-CEA-UGA-EMBL) within the Grenoble Partnership for Structural Biology (PSB), supported by FRISBI (ANR-10-INBS-0005-02) and GRAL, financed within the University Grenoble Alpes graduate school (Ecoles Universitaires de Recherche) CBH-EUR-GS (ANR-17-EURE-0003). The IBS Electron Microscope facility is supported by the Auvergne Rhône-Alpes Region, the Fonds Feder, the Fondation pour la Recherche Médicale and GIS-IBiSA. The IBS acknowledges integration into the Interdisciplinary Research Institute of Grenoble (IRIG, CEA). M.A. has been supported by MIUR - Ministero dell'Istruzione Ministero dell'Università e della Ricerca (Ministry of Education, University and Research) under the national project FSE/FESR - PON Ricerca e Innovazione 2014-2020 (N° AIM1887574, CUP: E18H19000350007). M.A. stay in IBS (Grenoble, France) was funded by Instruct-ERIC (proposal number: 16046), part of the European Strategy Forum on Research Infrastructures (ESFRI) and supported by national member subscriptions by the proposal APPID1673. We acknowledge OpenEye/Cadence for providing us with an academic license for the software used in these studies.

## Author contributions

V.Z.P. performed inhibitor design, synthesis, characterization, and wrote the manuscript; P.A.P. performed computer-aided inhibitor design studies; W.L.L, M.A. carried out negative staining TEM, G.E., M.A. performed cryo-EM data collections; M.A. carried out sample preparation for cryo-EM, data processing and wrote the manuscript; M.A., F.F., F.G., R.I., F.A. carried out X-ray crystallographic studies, refined and analyzed both X-ray- and cryo-EM-derived structural data; D.N. performed characterization of inhibitor reactivity with TGR; D.D.L., V.G., B.D. performed PK studies; J.J.J. supervised PK studies; L.N.M.H. performed photolabeling experiments; S.Y.A. performed ex vivo and in vivo studies, biochemical studies, and wrote the manuscript; R.P.L. preformed biochemical studies and wrote the manuscript; M.E.B., L.M.M. preformed biochemical studies; G.R.J.T. wrote the manuscript; F.A., D.L.W., P.A.P. wrote the manuscript and supervised all the research.

## Competing interests

Compounds in this manuscript are a part of US Provisional Application No. 63/392,214. Applicants: University of Illinois at Chicago, Chicago, Illinois, U.S.A.; Rush University Medical Center, Chicago, Illinois, U.S.A.; University of L'Aquila, L'Aquila, Italy. Inventors: P.A.P., D.L.W., F.A., V.Z.P., S.Y.A., M.A. COMPOSITIONS AND METHODS FOR TREATING SCHISTOSOMIASIS, 2022. There are no other competing interests to declare.
