## [Peer Review File · Nature Communications]

REVIEWER COMMENTS

Reviewer #1 (Remarks to the Author):

NCOMMS-22-44026-T

In the absence of a vaccine, global control of schistosomiasis is currently mediated by praziquantel monotherapy. As the authors have indicated, (over) reliance on a single drug (with described limitations including reduced in vivo activity against juvenile-stage worms) presents a substantial challenge in limiting the distribution and pathology of schistosomiasis should PZQ-insensitive schistosomes develop. Thus, new drugs (and drug targets) are urgently needed and justifiably argued as the driving force behind this extensive study.

Expanding upon their previous characterisation of small molecule fragments capable of binding to thioredoxin glutathione reductase (TGR), a validated drug target, the authors have developed analogues that display a novel mode of action (non-covalent inhibition) against *Schistosoma mansoni* TGR. The in vitro medicinal chemistry, biochemical and structural characterisation of these molecules appears to be thoroughly performed and the conclusions drawn are broadly supported by the data generated (caveat - I am neither a medicinal chemist nor a structural biologist). Importantly, some of the analogues described also may be applicable to the inhibition/modulation of TGR homologues expressed in pathogens responsible for malaria and elephantiasis.

However, the methodology (including the summaries/legend/footnotes associated with tables and figures) describing the ex vivo and in vivo experiments require substantial clarification and further justification; some of what is currently written affects my ability to interpret the results. I also question some of the authors' conclusions/interpretations being drawn from the results presented. While I can fully understand the authors' enthusiasm, the ex vivo/in vivo narrative needs to be tempered and the claims to be more evenly presented.

The ex vivo parasitological data in Table 1 is confusing as it currently stands. As this data presents the rationale for progressing into the in vivo studies, it needs to be edited for clarity. For example:

- 1) The SmTGR/HsTrxR1 'IC50' column is represented as μM . However, in some of the Table cells (i.e. 'slow' and 'fast' acting compounds), a percentage is provided. This leads to confusion (the superscript lettering defined in the footnote of the Table does not help). Please only include data in these Table cells that are represented by the unit of measurement in the column header. Create another column if necessary.
- 2) The ex vivo parasite 'LD50' phenotype column is expressed as μM at a 24 hr timepoint according to the header. However, this is only true for the *S. mansoni* adult worm and some of the NTS data (not

marked by *1-3). The *S. mansoni* NTS data (marked by *1-3) is derived from 72 hr and the *S. mansoni* 23 day (juvenile) as well as *S. japonicum* adult worm viability data is derived at 48 hr according to the Table superscript; confusingly, these assays were performed at 24 or 48 hr according to the Methods (line 794). So, it is difficult to follow the narrative. Therefore, please carefully edit this information so that the column descriptor/header applies to all of the data presented in every Table cell (and is harmonised with the Methods' section). This is important as one of the authors' conclusions (line 387) states that 'our compounds have similar activities against both *S. mansoni* and *S. japonicum* adult worms....'. As the data collected in this study seem to be derived from two different timepoints, post compound incubation (24 hr for adult *S. mansoni* and 48 hr for adult *S. japonicum*), this conclusion cannot be drawn. Either remove this sentence, temper it or provide data derived from the exact same timepoint.

The in vivo efficacy data also require further clarification, organisation, re-interpretation and potential further experimentation to fully justify the conclusions being drawn. For example:

1) Delivery of compounds for efficacy experiments was via the intraperitoneal route only. No compound was delivered via the oral route in an efficacy experiment. Therefore, it is unfair to say that 'novel druglike and orally bioavailable TGR inhibitors demonstrated efficacy against schistosome infections in mice' (lines 46-47) AND 'new druglike and orally bioavailable compounds identified here are active in animal models of schistosomiasis' (lines 459-460). While an oral PK experiment was performed with compound 2, the results of this did not inform or lead an efficacy study. Please rewrite these passages to avoid confusion.

2) Figure 4. The authors provide an 'image of liver of one mouse cured with 2' (Figure 4c). Without additional data to demonstrate that this mouse was actually infected (such as performing an anti-worm ELISA with sera collected from this mouse prior to treatment), the authors cannot say that it was cured. This statement and the designator in Figure 4a both need to be removed. However, if infection can be validated in this mouse (i.e. sera is available), then I also believe that the liver images in Figure 4 are incorrectly displayed. For example, the liver image in figure 4f (Compound 2; 1x100mg/kg) appears to be the one that does not contain egg-associated pathology (compared to Figure 4c; Compound 2; 1x100mg/kg).

3) Figure 4 and Extended Data Figure 7. For some experiments (e.g. Figure 4b; extended data figure 7a), the worm burdens obtained in each group (5 animals/group) are variable (as one would expect from a biological system). However, in some experiments (e.g. Figure 4e; extended data figure 7d and 7e), the worm burdens derived from 5 individual mice are invariable. How do the authors explain this?

4) The average control worm burdens across all efficacy experiments are quite variable as well (ranging from an average of ~11/mouse to ~38/mouse), which is a bit concerning when considering that all infections were initiated with 80 cercs. While some variability is expected (see above), I question whether this degree of inter-experiment variability in worm burdens derived from control groups affects power calculations (i.e. determining group size) and ultimately the ability to statistically identify efficacy?

5) It would appear that the data (efficacy of compounds 1 and 6 on mice infected three weeks previously) presented in extended Figures 7d and 7e are identical. Is this a mistake?

6) In most presentations of in vivo efficacy, the treated groups are labelled red and the control groups are labelled blue. Extended Data Figure 7c (compound 8) reverses the colours. Please correct to avoid confusion.

7) On several occasions, the authors make sweeping statements about their compounds' superior activity when compared to PZQ (e.g. lines 371-373, 'Novel TGR inhibitors are schistosomicidal ex vivo and outperform PZQ, the drug of choice to fight schistosomiasis, with superior activity against juvenile worms, which are less susceptible to PZQ treatments...'). The authors' conclusions are drawn from ex vivo data using a measurement of ATP production (as a surrogate for measuring viability), but the statement implies broader claims. This can be misleading without support derived from experiments directly comparing the in vivo effects of PZQ vs compounds 1 or 6 (such as those found in extended data Figures 7d and 7e) on mice previously infected three weeks earlier (harbouring juvenile worms). As only a 28% reduction in worm burden was achieved with compounds 1 or 6, how do the authors know that this is superior to what PZQ would achieve in a side by side comparison? Previous studies on 21-28 day *S. mansoni* infected mice demonstrated that PZQ has an efficacy ranging between 17-50% (DOI: 10.1007/BF00389899), which is within the range reported by the authors when assessing their tested compounds. Without performing in vivo comparative studies, the authors do not provide evidence supporting a superior role for their compounds over PZQ and need to temper their claims in the narrative.

The in vivo PK data also require further consideration by the authors. For example:

1) It is unclear why the three PK experiments (IP compound 1, 100mg/kg; IP compound 2, 100mg/kg; PO compound 2, 200 mg/kg) were performed after the efficacy studies (if, in fact, they were). Usually, these types of experiments are incredibly valuable in determining the predicted dosage required to reach a therapeutic threshold (i.e. reaching or surpassing the LD50 or LD90 obtained from ex vivo experiments; Table 1 for example) to plan efficacy experiments in the animal model used (here, the mouse). As none of the dosages or routes led to a compound concentration (at any timepoint measured; 30, 60 and 120 minutes) that reached/surpassed the LD50 found from ex vivo experiments on either the juvenile or adult lifecycle stages (Table 1), I am unsure how this data is being used. I also don't agree with the authors' statement in the discussion (lines 430-432) that 'in vivo plasma concentration of inhibitors 1 and 2 suggests that similarly to PZQ 2 hrs exposure at or above LD50 is sufficient to achieve strong schistosomicidal activity in vivo'. The authors' PK data do not support this statement (ex vivo LD50 compound 1 against adult *S. mansoni* = 12.3µM, Cmax of IP compound 1 = 4.1µM; ex vivo LD50 compound 2 against adult *S. mansoni* = 12.5µM, Cmax of IP compound 2 = 4.8µM, Cmax of PO compound 2 = 0.77µM). The authors need to reconsider how the PK data is being used and temper their discussion/claims accordingly.

Some minor comments that also require addressing:

1) Line 60 – there are new drugs in the (early stage) clinical pipeline for schistosomiasis (doi: 10.1371/journal.pntd.0009490). Perhaps this sentence could be updated?

2) Line 66 – paediatric PZQ formulations (oral dispensable tablets) are currently in clinical trials to improve the ability to treat children (DOI: 10.1371/journal.pntd.0007370). Perhaps including these types of advances in schistosomiasis control will more accurately reflect the state of the art?

3) The use of the word ‘Druglike’ on lines 367 and 459 (for examples). What do the authors actually mean? In other words, what data provided for their compounds support this definition or use of word? MW? logD?, solubility?, protein binding?, Ro5 characteristics?, etc.? I suggest that the authors support the use of this word with exemplars derived from data that they have collected.

4) The use of the word ‘toxicity’ on line 356. Again, what do the authors mean here? How was this determined/assessed? Do they actually mean ‘no adverse effects’? Defining these terms help the reader to understand what is meant.

5) The ‘Oral Gavage’ PK methodology on page 23 of the supplementary material/information. Please include the exact amount of compound 2 that was used (in the main body of the manuscript, a single dose of 200mg/kg is indicated).

6) Lines 378-379. Without performing a PK experiment that includes a tp at 24 hr (the ones conducted in this study stopped at 2 hr), how do the authors know that compound metabolism by the host detoxification systems would have occurred or been completed in a 24 hr timeframe?

Reviewer #2 (Remarks to the Author):

This study presents small molecule compounds that non-covalently bind to the thioredoxin glutathione reductase (TGR) of schistosomes and demonstrate very good schistosomicidal activity *ex vivo* and in mouse models. The study is significant in that there is growing resistance to the sole therapeutic on the market, praziquantel (PZQ). PZQ is also not efficacious against the juvenile form of the worm. This study offers a new direction, selectively drugging TGR via noncovalent targeting of a site adjacent to the NADPH binding site, the so-called “doorstop” (gatekeeper) pocket. The new compounds demonstrate activity against both the adult and juvenile worm.

There are a number of novel aspects to this work, not least in providing the first potent noncovalent inhibitors of TGR. The authors’ earlier work identified weaker noncovalently binding fragments from screening and crystallography which interact at the doorstop site (ACS Chem. Biol. 2018, 13, 2190–2202; ACS Infect. Dis. 2021, 7, 1932–1944). The current work successfully elaborates these fragments into somewhat related, larger druglike molecules, of low micromolar activity *in vitro* and importantly, good schistosomicidal activity *ex vivo* and *in vivo*. A cryo-EM structure, the first of a high molecular weight TrxR subfamily member, is presented, confirming the presence of one of the novel compounds in the doorstop pocket of TGR.

In terms of the methods employed, based on my expertise I focus on the molecular design aspects; the design approach appears logical and methods soundly employed, applying in silico tools to build on previous experimental fragment screening work. A subsequent array of experimental assays provide mechanistic insight and justify some of the resulting molecules as suitable leads for future compound optimisation. This is encouraging, given the need for new therapeutics in this area.

Therefore I recommend this novel contribution for publication in Nature Communications (subject to considering the following comments), as providing a novel and potentially therapeutically beneficial route to targeting schistosomiasis, a disease causing significant morbidity and mortality in subtropical and tropical populations.

The authors have previously identified covalent binder leads targeting TGR (*ACS Infect. Dis.* 2020, 6, 393–405). It would be useful to expand further on their discussion of the potential benefits of noncovalent leads for TGR in the Introduction and Discussion sections.

The methodology employed in compound design appears sound, building on the authors' earlier fragment screening/X-ray work. The authors state that an in-depth SAR analysis for the >100 compounds made in their campaign will be the subject of a subsequent publication, which seems reasonable given the focus on demonstrating the schistosomicidal efficacy of the lead compounds. However it would be useful to provide more detail in the main text on the function and deployment of the in silico tools to guide the design of compounds 1 – 8. For example, a design rationale is offered for the inclusion of the pinane ring, based on a hypothesis generated by the in silico design software GamePlan. It should be made clear that GamePlan generates a set of hypotheses for ligand modification based on relative stability calculations from probe molecule energetics computed using SZMAP, indicating regions favorable for polar and nonpolar ligand modifications. More detail on the approach used by SZMAP to compute energetics should be stated in Methods. Indeed, did SZMAP support their assumption that displacement of water in sites A and C would favor binding? (lines 112-115)

For the case of the pinane ring, nonpolar modifications were favored in the C site of the doorstep pocket. There is some confusion in the text, describing the doorstep as either a pocket (line 83) or as one of three subsites (line 116). This needs clarification.

The cryo-EM structure bears out the pinane group orientation, with matching electron density in the expected C site. The cryo-EM structure appears to be of adequate resolution to support the proposed ligand binding modes. It would be useful to indicate clearly on Fig 3c, the location of the C and other sites, as well as have a figure accompanying Extended Data Figure 1, indicating the pinane ring pose in C with the density around it. Given the ability of this and other compounds in the series to also bind well to human TrxR1 (Table 1), can this be understood in terms of analogous binding to its doorstep pocket/subsites?

The use of vBROOD is distinct from GamePlan/SZMAP, in offering suggestions for isosteric replacement of ligand core and peripheral functionality based on similar shape and electrostatics. This depends on a fragment library which was generated by CHOMP (this needs more clearly stated in Methods as “prepared using the CHOMP module to fragment the ChEMBL database”, line 890). How large was the ChEMBL fragment library?

Other comments:

What hydrophobic portions of residues D325 and H538 were engaged in site C? (line 105).

Was the optimization by MOE in the absence of solvent and was any significant structural distortion of the protein observed due to this? Make clear if this minimization is part of the “structure preparation” procedure. A reference to MOE should be inserted for Protonate3D (line 886).

Figure 1a – the doorstep pocket should be coloured differently from one of the TGR monomers for clarity.

“All compounds in Table 1” (line 376)

“thioredoxin-selective fluorescent probe” (line 158)

“NADPH-dependent reduction” (lines 409-410)

A phrase seems missing after “impressive” (line 437)

“synthesized” (line 207)

Reviewer #3 (Remarks to the Author):

Petukhova and colleagues present a substantive fragment guided design, synthesis and analysis in vitro and in situ of lead inhibitors targeting thioredoxin glutathione reductase in schistosome worms. The latter a vector of schistosomiasis, a broadly disseminated disease which plagues low income nations globally, the work will be of significant interest to the academic and pharmaceutical industry.

Some minor revisions are suggested:

Please clarify mass, (necessary dimeric, perfect or pseudo symmetry?) oligomerization, buried interface, any known allostery for TGR in the introduction. In prior xray crystallographic structures, are compounds typically found equivalently in both protomer active sites? Or differential binding?

For all tables/figures with error analysis, please clarify error treatments in the figure legend or footnote.

There are obviously many examples of powerful covalent inhibitors in the clinic, so perhaps this should be somewhat toned down in the introduction and elsewhere.

Line 247 etc “To increase aqueous solubility of the compounds to facilitate structural studies, the cycloheptyl substituent in TGR uncompetitive inhibitor 8 was replaced with a sugar moiety, resulting in compound 9, a noncompetitive inhibitor and thus capable of binding the enzyme in absence of NADP(H).” Were these modified compounds also tried for crystallography given the enhanced solubility? While nice to have the proof of principle of cryoEM, higher resolution obviously would have been a plus here.

For the cryoEM structure, C2 symmetry was applied in the data processing. Presumably therefore the compound was observed in both protomers equivalently (perhaps maps of both appropriate)? Can any assessments of local conformational changes be assessed that are induced by the binding of compound or is the local resolution/map too poor? It seems somewhat at the limits of the resolution here to assess the contributions of the alternate conformations observed, also coupled with the potential influence of the solubilizing sugar addition. Perhaps it is sufficient to conclude general localization given the data resolution, which is still an interesting and important contribution as per the unique site.

At the same time was it not possible to process in C1 to better clarify differences? Of course, enhanced particle collection on a higher energy microscope/faster detector could improve data quality.

Figure 3 could be made more clear. Is FAD observed or modelled in, please clarify in legend? (and provide density if the former?)

A ligplot could be more useful for the lay reader for general binding localizations (again with the caveat of resolution). Are there any potential hydrogen bond or electrostatic interactions? It appears largely hydrophobic? How do the authors believe specificity is being realized? A general discussion of specificity considerations in worms vs humans in this family of enzymes would be useful.

Validation report looks reasonable for something of this resolution, but perhaps a check of the clash score indications before final deposition would be prudent.

Reviewer #4 (Remarks to the Author):

Thioredoxin glutathione reductase (TGR), a selenoprotein, is essential for schistosome survival and therefore, can be a target for new drug development. Potassium antimonial tartrate and oltipraz were identified to inhibit TGR with irreversible and/or covalent binding, but their use was discontinued due to unacceptable side effects.

To develop non-covalent, metabolically stable, and druglike TGR inhibitors, the study obtained small molecule fragments by X-ray crystallography screening. It was ligated and partially optimized as a first-in-class non-covalent inhibitor of TGR. The authors clarified TGR cryo-EM structure and demonstrated that these inhibitors bind at a secondary site preventing the NADPH oxidation steps. The authors state that these novel druglike and orally bioavailable TGR inhibitors display schistosomicidal activity reaching the nanomolar range against different parasite-life cycle stages including juvenile worms and were effective against schistosome infections in mice.

This study shows new TGR inhibitors with non-covalent binding. The study's aim is of interest and importance, but there are many concerns and issues to be solved. 1) They inhibit humanTrxR1, indicating low selectivity, and raise questions about the safety profile, which can be a significant drawback. In addition, 2) the TGR inhibiting efficacies are not excellent, but they impair schistosomes with low concentrations. Thus, the cause/effect relationship remains to be clarified. 3) The authors state that the inhibitors bind to the ES complex, however, it is unclear which enzyme states the inhibitors bind to, oxidized form TGR/NADPH, reduced form TGR/NADP⁺, reduced form TGR/NADPH or reduced form TGR. 4) Based on the presented data, these TGR inhibitors do not appear to surpass the efficacy of PZQ on adult and egg burdens in vivo. 5) Although these inhibitors show some effect against juvenile worms, the reduction of egg burden does not seem to be better than that of PZQ. Therefore, statements in the abstract and discussion are not convincing in comparison with PZQ.

Title

It is rather difficult for readers who don't know TGR well to understand why the authors mention "non-covalent". Please explain the reason in the text more.

Results

Line 145: The decrease in GSH/GSSG ratio does not correlate with the inhibition potency of GTR. This leads to questions if the target/effect is really correlated to GTR or is caused by a non-specific effect due to the high concentration of the compounds tested in these experiments (50 μ M).

Line 158: It is recommended to describe that a selective fluorescent probe can work for imaging the enzyme activity.

Line 162: Fig. 2a

The experiment was done at 5 μM conc of compounds, which is a low concentration of the compounds where TGR is not inhibited.

Cpd 1, 42.1% inhibition at 67 μM ;

Cpd 2, 28.7% inhibition at 67 μM ;

Cpd 4, 68.6% inhibition at 67 μM ;

Cpd 7, $\text{IC}_{50} = 14.6 \mu\text{M}$;

Cpd 8, $\text{IC}_{50} = 10.3 \mu\text{M}$;

Why a decrease in fluorescence was observed at a concentration (5 μM) much lower than the concentration required to observe some inhibition to TGR according to the data from table 1?

Fig. 2c

The experiment was done at 30 μM conc of the compounds.

Line 164

From the extended data Fig 2a, b, c, it's hard to reach the conclusion that the novel TGR inhibitors engage TGR in ex vivo worms.

Line 174:

Please explain “the jump dilution assay” briefly to the readers.

Although the new TGR inhibitors found by the authors are reversible, these inhibitors lack selectivity as they also inhibit the mammalian TrxR1 and raise questions about the safety profile.

Line 194

Why the inhibition of TGR is NADPH dependent? The cryo-EM structures of TGR with the non-covalent inhibitors were determined even in the absence of NADPH. If the non-covalent inhibitors cannot bind to

the TGR in absence of NADPH, does this mean that the cryo-EM structures described by the authors represent non-physiological structures?

Line 206:

K, V and T of T_m should be written in italic.

Line 210:

"poor" is not scientific word. I recommend editing the following descriptions used in other paragraphs, such as "our compound" or "better".

Line 219:

At 500 μM of NADPH, can the reduced form of the enzyme also bind to NADPH, similar to what has been observed to the yeast type 2 NADH dehydrogenases (in this case NADH binds to the reduced form of the enzyme)?

Also, in the PRP labelling, there is labelling in the TGR in the absence of NADPH, although at a less effective rate.

Therefore, would you explain why the authors believe that these inhibitors bind to the ES complex and not to the reduced form of the enzyme?

Why the compound 9 (a noncompetitive inhibitor and thus capable of binding the enzyme in the absence of NADP(H)) was not included in the TSA?

Line 242:

The authors explain the result of low solubility not only here but also later part. How have the authors tried the suitable conditions? More information on the trial should be described.

Line 278:

A description "generally" is not clear.

Line 284:

Result on the enzyme from the Vero cell and the effect of the compounds on human cells should be described.

Line 287:

Why do the authors describe the effect of cpd 6 on TrxR1 as "no appreciable inhibition" whilst as "significant inhibition" of cpd 2 on TGR, even if the degree of inhibition at 67 μ M for cpd 6/TrxR1 (28.9%) pairs and cpd 2/TGR (28.7%) are very close?

Line 292:

Again, at such low concentrations, it doesn't seem to inhibit TGR.

As mentioned before, "better" is not a scientific word.

Line 302:

To what extent the structures of TGR from Sm and Sj are similar or different?

Can the effect on Sj be explained by, for example, its TGR modelled structure (Alpha fold model or any other model with high confidence)?

Line 305:

"good" is not a scientific word.

From lines 320~333:

This doesn't seem to be potent. Please include the data or compare them with the positive control group treated with PZQ.

From lines 334~339:

The activity against juveniles doesn't seem to be potent as well.

Why the authors did not investigate the effect of these compounds against schistosomula recovered from the lungs, as most of them seem to be more active to the NTS than in adults, as expected from Table 1.

Line 350:

Will cpd 1 inhibit TGR at this plasma concentration (4.1 μM)?

Line 353:

Will cpd 2 inhibit TGR at this plasma concentration (4.8, 3.9, and 3.0 μM)?

Line 356:

The toxicity study result should be shown.

Discussion

Line 440~443:

From the data presented by the authors, the in vivo effect of these TGR inhibitors does not appear to surpass the efficacy of PZQ to adult and egg burdens.

Although these inhibitors have some effect against juvenile worms, the reduction of egg burden does not seem to be better than that of PZQ.

Therefore, such a claim does not seem to be convincing when compared with PZQ.

Line 454~457

Can the lack of specificity of the presented inhibitors between TGR and human TrxR1 be explained by their structures (determined or modelled)?conclusions.

The reviewers comments to be addressed are in “*italic*”. Our responses are in regular font in blue.

Reviewer #1

The ex vivo parasitological data in Table 1 is confusing as it currently stands. As this data presents the rationale for progressing into the in vivo studies, it needs to be edited for clarity. For example:

1) The SmTGR/HsTrxR1 'IC50' column is represented as μM . However, in some of the Table cells (i.e. 'slow' and 'fast' acting compounds), a percentage is provided. This leads to confusion (the superscript lettering defined in the footnote of the Table does not help). Please only include data in these Table cells that are represented by the unit of measurement in the column header. Create another column if necessary.

We have broken table 1 into 2 tables as suggested.

*2) The ex vivo parasite 'LD50' phenotype column is expressed as μM at a 24 hr timepoint according to the header. However, this is only true for the *S. mansoni* adult worm and some of the NTS data (not marked by *1-3). The *S. mansoni* NTS data (marked by *1-3) is derived from 72 hr and the *S. mansoni* 23 day (juvenile) as well as *S. japonicum* adult worm viability data is derived at 48 hr according to the Table superscript; confusingly, these assays were performed at 24 or 48 hr according to the Methods (line 794). So, it is difficult to follow the narrative. Therefore, please carefully edit this information so that the column descriptor/header applies to all of the data presented in every Table cell (and is harmonised with the Methods' section). This is important as one of the authors' conclusions (line 387) states that 'our compounds have similar activities against both *S. mansoni* and *S. japonicum* adult worms....'. As the data collected in this study seem to be derived from two different timepoints, post compound incubation (24 hr for adult *S. mansoni* and 48 hr for adult *S. japonicum*), this conclusion cannot be drawn. Either remove this sentence, temper it or provide data derived from the exact same timepoint.*

Reconstructed tables. The *1-3 were mistakenly placed within the table and have been removed. These should only apply to the PRP in the footnotes.

The in vivo efficacy data also require further clarification, organisation, re-interpretation and potential further experimentation to fully justify the conclusions being drawn. For example:

1) Delivery of compounds for efficacy experiments was via the intraperitoneal route only. No compound was delivered via the oral route in an efficacy experiment. Therefore, it is unfair to say that 'novel druglike and orally bioavailable TGR inhibitors demonstrated efficacy against schistosome infections in mice' (lines 46-47) AND 'new druglike and orally bioavailable compounds identified here are active in animal models of schistosomiasis' (lines 459-460). While an oral PK experiment was performed with compound 2, the results of this did not inform or lead an efficacy study. Please rewrite these passages to avoid confusion.

Oral bioavailability was evidenced by PK studies. These statements have been rewritten to avoid confusion.

2) *Figure 4. The authors provide an ‘image of liver of one mouse cured with 2’ (Figure 4c). Without additional data to demonstrate that this mouse was actually infected (such as performing an anti-worm ELISA with sera collected from this mouse prior to treatment), the authors cannot say that it was cured. This statement and the designator in Figure 4a both need to be removed. However, if infection can be validated in this mouse (i.e. sera is available), then I also believe that the liver images in Figure 4 are incorrectly displayed. For example, the liver image in figure 4f (Compound 2; 1x100mg/kg) appears to be the one that does not contain egg-associated pathology (compared to Figure 4c; Compound 2; 1x100mg/kg).*

Supplementary Note I has egg counts (eggs/gram liver tissue) from livers of treated mice. For each treatment/control pair, the infection is done with mice acquired from the same source at the same time, with cercariae obtained from the same batch of snails. Both the infection and perfusion are blinded. Treatments directed at adult worms commenced 6 weeks after infection, when the worms were mature, paired, and producing eggs. It is unlikely worms die immediately from the treatment and, therefore, continue to produce eggs until death. The livers from each mouse that were treated have eggs, indicating that the treated mice were infected. We argue that it is not necessary to perform ELISAs to verify infection; infections are verified by the presence of eggs.

The images in Figure 4 (now Fig. 6) are correctly displayed. The liver shown in 6c is from treatment targeting adult worms. Treatments directed at adult worms commenced 6 weeks after infection, when the worms are mature, paired, and producing eggs. It is unlikely worms die immediately from the treatment and, therefore, continue to produce eggs until death. Fig. 6f shows treatment targeting juvenile, immature worms, before egg production commences. This image shows only one granuloma. However, the images are not meant to be quantitative, only descriptive. Detailed egg counts for each experiment are shown in the figures.

3) *Figure 4 and Extended Data Figure 7. For some experiments (e.g. Figure 4b; extended data figure 7a), the worm burdens obtained in each group (5 animals/group) are variable (as one would expect from a biological system). However, in some experiments (e.g. Figure 4e; extended data figure 7d and 7e), the worm burdens derived from 5 individual mice are invariable. How do the authors explain this?*

Mice are age-matched and cercariae are derived from the same snails for each drug treatment. Activity of cercariae can vary from experiment to experiment, with some snail sheds producing highly active and infective cercariae and others producing more variably active cercariae. This is dependent on many factors, for example, the time after snails were infected to collect cercariae and the intensity of snail infections. Although the mice are age-matched in a given experiment, different experiments used mice of slightly different ages and as mice age the tail skin thickness

and infectability can become more variable. These are possible reasons why there is variation in control infection intensity from experiment.

4) *The average control worm burdens across all efficacy experiments are quite variable as well (ranging from an average of ~11/mouse to ~38/mouse), which is a bit concerning when considering that all infections were initiated with 80 cercs. While some variability is expected (see above), I question whether this degree of inter-experiment variability in worm burdens derived from control groups affects power calculations (i.e. determining group size) and ultimately the ability to statistically identify efficacy?*

We agree that the average control worm burdens across the experiments are quite variable. This is attributed to several factors such as the age/batch of mice and snails used for the exposure. As mice age, their tail skin becomes thicker reducing infection efficiency. As snails/sporocysts age, the cercariae produced will have different infection efficiency. Nevertheless, we estimated an experimental power at 80% alpha, to control for any such confounders. Moreover, inference for the efficacy experiments were made based on differences between treated and untreated groups in the same experiment, in which all mice and snails were of the same age, and not between different experimental set ups.

5) *It would appear that the data (efficacy of compounds 1 and 6 on mice infected three weeks previously) presented in extended Figures 7d and 7e are identical. Is this a mistake?*

A mistake. The figure has been corrected. It is now in Supplemental Notes.

6) *In most presentations of in vivo efficacy, the treated groups are labelled red and the control groups are labelled blue. Extended Data Figure 7c (compound 8) reverses the colours. Please correct to avoid confusion.*

The figure has been corrected. It is now in Supplemental Notes.

7) *On several occasions, the authors make sweeping statements about their compounds' superior activity when compared to PZQ (e.g. lines 371-373, 'Novel TGR inhibitors are schistosomicidal ex vivo and outperform PZQ, the drug of choice to fight schistosomiasis, with superior activity against juvenile worms, which are less susceptible to PZQ treatments...'). The authors' conclusions are drawn from ex vivo data using a measurement of ATP production (as a surrogate for measuring viability), but the statement implies broader claims. This can be misleading without support derived from experiments directly comparing the in vivo effects of PZQ vs compounds 1 or 6 (such as those found in extended data Figures 7d and 7e) on mice previously infected three weeks earlier (harbouring juvenile worms). As only a 28% reduction in worm burden was achieved with compounds 1 or 6, how do the authors know that this is superior to what PZQ would achieve in a side by side comparison? Previous studies on 21-28 day *S. mansoni* infected mice demonstrated that PZQ has an efficacy ranging between 17-50% (DOI: 10.1007/BF00389899), which is within the range reported by the authors when assessing their tested compounds. Without performing in vivo comparative studies, the authors do not provide*

evidence supporting a superior role for their compounds over PZQ and need to temper their claims in the narrative.

In vivo activity. We tested compounds at day 23 p.i. Treatment with compound **2** at 1 X 100 mg/kg resulted in >60% reduction in worm burdens. In the referenced paper, treatments with PZQ at 1 X 500 mg/kg resulted in 0% reductions at 21 and 28 days p.i. and treatments at 1 X 1000 mg/kg resulted in 50 % and 17% at day 21 and 28 p.i., respectively. In reference 43, treatments targeting 28 day old worms resulted in ED₅₀ = 2456 mg/kg. The ED₅₀ for compound **2** is < 100 mg/kg. On this basis, we conclude that compound **2** has superior *in vivo* activity to PZQ.

Ex vivo activity. An LD₅₀ = 413 μM for PZQ against juveniles was previously reported (PMID: 15013742). We found that PZQ had LD₅₀ >50 μM (the highest concentration tested), while compounds **1-5** had LD₅₀s between 7.2 and 26 μM. Against *ex vivo* juvenile worms, compounds **1-5** have superior activity to PZQ against *ex vivo* worms.

Text has been changed to focus on compound **2** in *in vivo* studies and to stress superior *ex vivo* activity of the compounds **1-5**.

The in vivo PK data also require further consideration by the authors. For example:

1) It is unclear why the three PK experiments (IP compound 1, 100mg/kg; IP compound 2, 100mg/kg; PO compound 2, 200 mg/kg) were performed after the efficacy studies (if, in fact, they were).

We agree with the reviewer but would like to highlight that this is true for an ideal drug discovery process, and it is more typical for more advanced stages of drug development after the potential for further development was established. PK/PD studies and especially PK/PD studies with multiple time points and multiple doses are prohibitively expensive and, typically, done after a promising lead series is selected.

Furthermore, this manuscript is a proof-of-concept study, a first report of non-covalent TGR inhibitors with *in vivo* activity. Hence, the PK studies were performed after we obtained a proof-of-concept for efficacy of these compounds *in vivo*. Similarly, the PK studies using an oral dose were performed to understand if this series of compounds has a potential to be administered orally. The efficacy data and the PK data obtained IP and PO provide a very strong rationale for further drug discovery efforts. Running additional PK studies and additional efficacy studies for compounds that are currently optimized would provide little to no benefit.

*Usually, these types of experiments are incredibly valuable in determining the predicted dosage required to reach a **therapeutic threshold** (i.e. reaching or surpassing the LD50 or LD90 obtained from *ex vivo* experiments; Table 1 for example) to plan efficacy experiments in the animal model used (here, the mouse). As none of the dosages or routes led to a compound concentration (at any timepoint measured; 30, 60 and 120 minutes) that reached/surpassed the LD50 found from *ex vivo* experiments on either the juvenile or adult lifecycle stages (Table 1), I am unsure how this data is being used.*

As we mentioned above, this is a proof-of-concept study, not a full-blown drug development undertaking. Additional research is underway to find more potent and more efficacious compounds and understand PK/PD relationship *in vivo*, especially as it relates to LD₅₀ determined *ex vivo*.

We also would like to point out that although the reviewer's statement is generally and conceptually correct, it is based on the assumption that LD₅₀ values determined *ex vivo* are accurate estimates of the therapeutic threshold *in vivo*. There are many different ways to determine LD₅₀ *ex vivo* (discussed in the paper), and the field is not settled on just one. Most methods rely on treatment and observation for up to 8 days and somewhat subjective phenotypic scoring system. Moreover, because all the aspects of the host biology are missing in the culture *ex vivo*, they are not factored in the LD₅₀ determined *ex vivo*. The most obvious example is PZQ. In clinical use in patients, PZQ works at sub LD₅₀ concentrations, which were determined *ex vivo*. Most likely, treatment with PZQ "unmasks" the worm to the immune defenses of the host, and this phenomenon was reported in the literature. There are hundreds of publications exploring the "unmasking" phenomenon in the anti-infective and other therapeutic areas.

Worms in an animal host are under higher stress than cultured, *ex vivo* worms. For example, the host immune system generates reactive oxygen species targeting worms, a stress offset by TGR activity. This stress is not present for cultured worms. Therefore, there may not be a direct correlation between compound activity when tested against *in vivo* and *ex vivo* worms. That is, compounds may have greater activity against worms in mice than against cultured worms. We have shown this with silencing of peroxiredoxin 1 (Prx1), an enzyme downstream of TGR, and challenging worms with H₂O₂, normally detoxified by Prx1 (Trx and/or GSH dependent) in worms (worms do not have catalase) (PMID: 16606626). We treated NTS with 100 μM or 200 μM H₂O₂ and monitored worm survival. Worms without Prx1 silencing survived H₂O₂ challenge equal to unchallenged worms, but silencing Prx1 led to significantly increased death when exposed to both H₂O₂ challenges compared to non-silenced worms. Worms are under greater stress in a host so that a compound may be more active against *in vivo* worms than against *ex vivo* worms.

Additional explanation/details have been added to the discussion section to address the reviewer's concerns.

I also don't agree with the authors' statement in the discussion (lines 430-432) that 'in vivo plasma concentration of inhibitors 1 and 2 suggests that similarly to PZQ 2 hrs exposure at or above LD50 is sufficient to achieve strong schistosomicidal activity in vivo'. The authors' PK data do not support this statement (ex vivo LD50 compound 1 against adult S. mansoni = 12.3 μM, Cmax of IP compound 1 = 4.1 μM; ex vivo LD50 compound 2 against adult S. mansoni = 12.5 μM, Cmax of IP compound 2 = 4.8 μM, Cmax of PO compound 2 = 0.77 μM). The authors need to reconsider how the PK data is being used and temper their discussion/claims accordingly.

This sentence has been removed from the text.

Some minor comments that also require addressing:

1) Line 60 – there are new drugs in the (early stage) clinical pipeline for schistosomiasis (doi: 10.1371/journal.pntd.0009490). Perhaps this sentence could be updated?

The text has been modified to reflect this publication.

2) Line 66 – paediatric PZQ formulations (oral dispensable tablets) are currently in clinical trials to improve the ability to treat children (DOI: 10.1371/journal.pntd.0007370). Perhaps including these types of advances in schistosomiasis control will more accurately reflect the state of the art?

The text has been modified to reflect this publication.

3) The use of the word ‘Druglike’ on lines 367 and 459 (for examples). What do the authors actually mean? In other words, what data provided for their compounds support this definition or use of word? MW? logD?, solubility?, protein binding?, Ro5 characteristics?, etc.? I suggest that the authors support the use of this word with exemplars derived from data that they have collected.

In the original submission, the Lipinski *et al.* druglikeness/oral bioavailability criteria were provided in the supplemental section S1 and referenced in the main text. The main text and the supplemental information were updated with an additional reference to Lipinski *et al.* publication. All the compounds and their MW, logP, and the number of donors and acceptors (as used in Lipinski *et al.* Ro5) are in the Supplemental Information.

4) The use of the word ‘toxicity’ on line 356. Again, what do the authors mean here? How was this determined/assessed? Do they actually mean ‘no adverse effects’? Defining these terms help the reader to understand what is meant.

Changed “toxicity” to “visible adverse effects”.

5) The ‘Oral Gavage’ PK methodology on page 23 of the supplementary material/information. Please include the exact amount of compound 2 that was used (in the main body of the manuscript, a single dose of 200mg/kg is indicated).

The PK methodology section was updated to include the dose of compound 2 as requested.

6) Lines 378-379. Without performing a PK experiment that includes a tp at 24 hr (the ones conducted in this study stopped at 2 hr), how do the authors know that compound metabolism by the host detoxification systems would have occurred or been completed in a 24 hr timeframe?

We were not implying this at all. We were only comparing our study to earlier studies that exposed worms for much more than 1 day or exposed for 1 day and waited many days to assess worm viability. To avoid any confusion, the statement has been removed.

Reviewer #2

The authors have previously identified covalent binder leads targeting TGR (ACS Infect. Dis. 2020, 6, 393–405). It would be useful to expand further on their discussion of the potential benefits of noncovalent leads for TGR in the Introduction and Discussion sections.

A discussion about the importance of identifying non-covalent inhibitors for TrxRs/TGRs has been added in the Introduction section.

The methodology employed in compound design appears sound, building on the authors' earlier fragment screening/X-ray work. The authors state that an in-depth SAR analysis for the >100 compounds made in their campaign will be the subject of a subsequent publication, which seems reasonable given the focus on demonstrating the schistosomicidal efficacy of the lead compounds. However it would be useful to provide more detail in the main text on the function and deployment of the in silico tools to guide the design of compounds 1 – 8. For example, a design rationale is offered for the inclusion of the pinane ring, based on a hypothesis generated by the in silico design software GamePlan.

Additional details on how the *in silico* tools were deployed were added to the beginning of the Results section of the manuscript.

It should be made clear that GamePlan generates a set of hypotheses for ligand modification based on relative stability calculations from probe molecule energetics computed using SZMAP, indicating regions favorable for polar and nonpolar ligand modifications. More detail on the approach used by SZMAP to compute energetics should be stated in Methods. Indeed, did SZMAP support their assumption that displacement of water in sites A and C would favor binding? (lines 112-115)

The difference between the outcomes of GamePlan and SZMAP calculations were clarified in the text. The results and methods sections were expanded. Yes, SZMAP calculations support displacement of water from the hydrophobic subpocket.

For the case of the pinane ring, nonpolar modifications were favored in the C site of the doorstep pocket. There is some confusion in the text, describing the doorstep as either a pocket (line 83) or as one of three subsites (line 116). This needs clarified.

This is now clarified at the beginning of the Results section “Fragment-based drug design and chemistry”.

The cryo-EM structure bears out the pinane group orientation, with matching electron density in the expected C site. The cryo-EM structure appears to be of adequate resolution to support the proposed ligand binding modes. It would be useful to indicate clearly on Fig 3c, the location of the C and other sites, as well as have a figure accompanying Extended Data Figure 1, indicating the pinane ring pose in C with the density around it. Given the ability of this and other compounds in the series to also bind well to human TrxR1 (Table 1), can this be understood in terms of analogous binding to its doorstep pocket/subsites?

The locations of subpocket C and of the other subsites, relative to the cryo-EM density of compound **9**, are now indicated in the new version of Figure 4. A magnification of the cryo-EM density relative to the pinane group in subpocket C and its surrounding residues is now present in the Supplementary Note J.

We feel confident to discuss data on compound **9**, the one which experimental structural data are available. Most of the residues that contact **9** in TGR are conserved in hTrxR1, except the three residues G437, P440 and Q441 that in hTrxR1 are replaced by an aspartate, a valine and a glutamate, respectively (see Supplementary Note K). However, as judging by cryo-EM density, these residues are implicated in the binding of the mobile sugar moiety only with their main chains. These contacts are supposed to contribute little to the binding free energy of the compound as **9** is found in double conformation due to an alternative position of its soluble sugar moiety (see figure 3). This is consistent with the similar inhibition potency of **9** against TGR and hTrxR ($IC_{50} = 57.5 \mu\text{M}$ against TGR; $IC_{50} = 70.5 \mu\text{M}$ against hTrxR, see new Table1).

The use of vBROOD is distinct from GamePlan/SZMAP, in offering suggestions for isosteric replacement of ligand core and peripheral functionality based on similar shape and electrostatics. This depends on a fragment library which was generated by CHOMP (this needs more clearly stated in Methods as “prepared using the CHOMP module to fragment the ChEMBL database”, line 890). How large was the ChEMBL fragment library?

The total number of fragments in the resulting database was 17,697,078. We amended the text as suggested and clarified the parameters for the CHOMP software.

Other comments:

What hydrophobic portions of residues D325 and H538 were engaged in site C? (line 105).

CH₂ portions of D325 and H538 and carbon atoms in the imidazole ring of H538 are hydrophobic. The polar groups in these residues point away from subpocket C.

Was the optimization by MOE in the absence of solvent and was any significant structural distortion of the protein observed due to this?

The methods section was amended to address this concern. The co-crystallized water molecules were kept in the binding site. The R-field option was used for the solvent model during minimization. No or very insignificant changes were observed in the structure of both the TGR protein and the residues in proximity to the small molecule fragments. The RMSD (CA) was c.a. 0.8 angstrom for the 24 residues located within 4.5 angstrom of the fragments, whereas the RMSD for the complete protein was c.a. 0.9 angstrom.

The RMSD values between the models minimized with the “R-field” and “Born” options for solvent were 0.7 and 0.26 angstrom for the whole protein and the binding site residues, respectively. Inclusion of only CA or all heavy atoms in the protein had little effect on the alignment and RMSD values.

Make clear if this minimization is part of the “structure preparation” procedure. A reference to MOE should be inserted for Protonate3D (line 886).

The methods section was amended to clarify the minimization and protonation procedures. Labute 2009 reference was inserted for Protonate3D.

Figure 1a – the doorstep pocket should be coloured differently from one of the TGR monomers for clarity.

The surface of the doorstep pocket is now colored differently in Figure 1a.

“All compounds in Table 1” (line 376)

Corrected

“thioredoxin-selective fluorescent probe” (line 158)

Corrected. This is actually a thioredoxin reductase-selective probe.

“NADPH-dependent reduction” (lines 409-410)

Corrected

A phrase seems missing after “impressive” (line 437)

Corrected

“synthesized” (line 207)

Corrected

Reviewer #3

Please clarify mass, (necessary dimeric, perfect or pseudo symmetry?) oligomerization, buried interface, any known allostery for TGR in the introduction. In prior xray crystallographic structures, are compounds typically found equivalently in both protomer active sites? Or differential binding?

TGR is a 130 kDA obligate homodimer as the functional stereochemistry of the FAD redox site in each subunit is generated by protein dimerization. This information and others details pertinent to the mechanism of action are now added in the Introduction.

To the best of our knowledge allosteric regulation has not been reported for TrxRs/TGRs, even though possible. Most of the TGR X-ray structures (15) with ligands/inhibitors, also bound in the doorstep pocket, have been solved by us in the C2 space group with one subunit of the dimer in the asymmetric unit, indicating that ligand binding at the doorstep pocket does not trigger allosteric effects between the subunits, at least in crystals.

For all tables/figures with error analysis, please clarify error treatments in the figure legend or footnote.

The information has been provided as suggested.

There are obviously many examples of powerful covalent inhibitors in the clinic, so perhaps this should be somewhat toned down in the introduction and elsewhere.

This has been toned down in the Introduction section. We expanded this section and now provide a short but fair and balanced analysis of advantages and disadvantages of the covalent inhibitors.

Line 247 etc “To increase aqueous solubility of the compounds to facilitate structural studies, the cycloheptyl substituent in TGR uncompetitive inhibitor 8 was replaced with a sugar moiety, resulting in compound 9, a noncompetitive inhibitor and thus capable of binding the enzyme in absence of NADP(H).” Were these modified compounds also tried for crystallography given the enhanced solubility? While nice to have the proof of principle of cryoEM, higher resolution obviously would have been a plus here.

All compounds in this study were tried in a first round in X-ray experiments. Compound 9 was optimized for solubility for X-crystallography studies but any attempts to obtain co-crystal structures with TGR (with or without NADPH, in different crystallization conditions and at different compound concentrations) were unsuccessful. Therefore, we switched to cryo-EM. This is now stated more clearly in the main text.

For the cryoEM structure, C2 symmetry was applied in the data processing. Presumably therefore the compound was observed in both protomers equivalently (perhaps maps of both appropriate)? Can any assessments of local conformational changes be assessed that are induced by the binding of compound or is the local resolution/map too poor? It seems somewhat at the limits of the resolution here to assess the contributions of the alternate conformations observed, also coupled with the potential influence of the solubilizing sugar addition. Perhaps it is sufficient to conclude general localization given the data resolution, which is still an interesting and important contribution as per the unique site.

Yes, the compound was detected in both subunits equivalently and is now shown in the new Figure 4. We do not see at this resolution (3.5 Å is the local resolution at the doorstep pocket) evident conformational changes induced by compound binding. We agree with the reviewer that (i) the structure is unique and important as it tells us that the adopted drug design strategy is successful and (ii) the resolution limits what we can say about the details of the interactions between 9 and the amino acidic residues of TGR. However, the cryo-EM map, even at low resolution, clearly indicates the presence of two poses of 9 that are entirely compatible with the chemical environment of the binding site (see new Figure 4).

At the same time was it not possible to process in C1 to better clarify differences? Of course, enhanced particle collection on a higher energy microscope/faster detector could improve data quality.

In C1, resolution becomes worst (3.9-4.0 Å) together with the correlation coefficients and in the general appearance of the cryo-EM density after structural refinement. Instead, we preferred to

process with the C2 symmetry because it resulted in better resolution (this information is now reported in the Methods section). In light of the intriguing possibility that ligands bound at the doorstep may induce some sort of asymmetry in the dimer, we are going to request beam time at the 300kV cryo-EM facilities for higher resolution data collections, trying other inhibitors also.

Figure 3 could be made more clear. Is FAD observed or modelled in, please clarify in legend? (and provide density if the former?)

Figure 4 is now completely remade and the cryo-EM densities for both the compound and the observed FAD are present.

A ligplot could be more useful for the lay reader for general binding localizations (again with the caveat of resolution). Are there any potential hydrogen bond or electrostatic interactions? It appears largely hydrophobic? How do the authors believe specificity is being realized? A general discussion of specificity considerations in worms vs humans in this family of enzymes would be useful.

2D representations of the ligand contacts are added to the new Figure 4 for both conformations. The hydrogen bonds are probably formed between the two conformers of the sugar moiety with the main chain carbonyls, in the text we refer to these as polar interactions since at this resolution we cannot be sure about the actual orientations of the involved atoms. However, the interaction surface between **9** and the doorstep pocket appears largely hydrophobic. Compound **9** does not present preferential inhibition of TGR over human TrxR ($IC_{50} = 57.6 \mu\text{M}$ for TGR and $70.5 \mu\text{M}$ for hTrxR). This can be rationalized by the binding pose found in the cryo-EM structure as most residues that contact **9** are conserved in both enzymes. Only three residues (G437, P440 and Q441) that bind the mobile sugar moiety are different in hTrxR, but they contribute to binding of **9** only with their backbone. Structural differences between the human and the worm enzyme are now reported in the Discussion, suggesting that selective inhibition of TGR over human TrxR is attainable. As a matter of fact, compound **6** has $IC_{50} = 2.5 \mu\text{M}$ against TGR and against human TrxR it is 28.9% inhibition at $66.7 \mu\text{M}$ (the highest concentration tested, see Table 1), so at least 25X more potent against TGR than human TrxR (See Discussion).

We are planning to address this very important aspect of selectivity in future optimization of these unoptimized lead compounds, collecting high resolution cryo-EM data on different inhibitors in complex with TGR and human TrxR1. An additional consideration is that humans have GR and worms do not. None of the compounds had any GR inhibition. At least in the short term, the GSH pathway can compensate for many of the functions of the Trx pathway. While in worms both the Trx and GSH pathways are affected when TGR is inhibited, in humans only the Trx pathway would be affected by TrxR inhibition.

Validation report looks reasonable for something of this resolution, but perhaps a check of the clash score indications before final deposition would be prudent.

We spent a lot of time trying to adjust the clashes manually as at this resolution the refining programs at each refining cycle continuously introduce them, even though we keep the geometry very tight during structural refinement. The clashes present in the pdb are below 0.76 \AA and do not involve the ligand and its binding site.

Reviewer #4

This study shows new TGR inhibitors with non-covalent binding. The study's aim is of interest and importance, but there are many concerns and issues to be solved. 1) They inhibit human TrxR1, indicating low selectivity, and raise questions about the safety profile, which can be a significant drawback.

We are aware that inhibition of human TrxR could affect the safety profile. It should be noted that these compounds are expected to work with a single dose, hence, inhibition of any off-targets in the host will be short-term. Moreover, the compounds described in this study are not yet optimized for human studies; this will be a goal of future work. The data presented here indicate that preferential inhibition of TGR over human TrxR is possible. Data in Table 1 show that compound **6** has $IC_{50} = 2.5 \mu\text{M}$ against TGR, against human TrxR it is $> 66.7 \mu\text{M}$ (the highest concentration tested), at least 25X more potent against TGR than human TrxR. Furthermore, human GR is spared from inhibition by all compounds shown. The human redox network is redundant, with separate Trx and GSH systems with redundancy and compensatory activities so that inhibition of one arm can be supplemented by the other.

2) the TGR inhibiting efficacies are not excellent, but they impair schistosomes with low concentrations. Thus, the cause/effect relationship remains to be clarified.

Interpretation of compound activity against recombinant TGR and against TGR in worms is not straightforward as we extensively discuss in the manuscript. We performed orthogonal assays - GSH/GSSG ratio and TRFS-Green - to determine that TGR inhibition in worms occurs and is linked to worm killing. We showed that TGR in worms was inhibited using a cell permeable TrxR/TGR probe/substrate TRFS-green. Robust inhibition of TGR in worms was observed at $5 \mu\text{M}$ using the TRFS-Green assay. Second, we determined the GSH/GSSG ratio in worms. Similarly to PZQ, schistosomicidal activity can be amplified by the host immune response *in vivo*. We added a pertinent discussion to the Discussion part of the manuscript.

3) The authors state that the inhibitors bind to the ES complex, however, it is unclear which enzyme states the inhibitors bind to, oxidized form TGR/NADPH, reduced form TGR/NADP⁺, reduced form TGR/NADPH or reduced form TGR.

Thank you for pointing out this. We realized that referring to the reduced species as NADP(H)-TGR can create confusion and so in the revised version we refer to these as NADP⁺-TGR(H) forms, indicating that the electrons are inside the polypeptide chain of TGR. Based on the structural and functional data we collected in the present and previous studies, we provide a hypothesis of the mechanism of the inhibitors in Figure 5 where we propose that the inhibitors prevent the release of NADP⁺ from the active site of the EH₂ species, slowing down binding of a second NADPH and formation of the EH₄, which is the one competent for catalysis. EH₂ and

EH₄ are the two reduced species formed upon binding of 1 and 2 molecules of NADPH, respectively. This is in line with the fact that uncompetitive inhibitors can exert their action by preventing product release (in our case NADP⁺) from the active site (doi: 10.1038/nrd.2017.219). Please, see our answers below and the Results section “TGR inhibitors target a reduced form of TGR”.

4) Based on the presented data, these TGR inhibitors do not appear to surpass the efficacy of PZQ on adult and egg burdens in vivo.

Our compounds are not yet fully optimized but have activity similar to PZQ *in vivo*. The major advantage of these unoptimized compounds is their superior activity against juvenile liver stage worms.

In vivo activity. We tested compounds at day 23 p.i. Treatment with compound **2** at 1 X 100 mg/kg resulted in >60% reduction in worm burdens. In the referenced paper, treatments with PZQ at 1 X 500 mg/kg resulted in 0% reductions at 21 and 28 days p.i. and treatments at 1 X 1000 mg/kg resulted in 50 % and 17% at day 21 and 28 p.i., respectively. In PMID: 15013742, PZQ treatments targeting 28 day old worms resulted in ED₅₀ = 2456 mg/kg. The ED₅₀ for compound **2** is < 100 mg/kg. On this basis, we conclude that compound **2** has superior *in vivo* activity to PZQ.

Ex vivo activity. An LD₅₀ = 413 μM for PZQ against juveniles was previously reported (PMID: 15013742). We found that PZQ had LD₅₀ >50 μM (the highest concentration tested), while compounds **1-5** had LD₅₀s between 7.2 and 26 μM. Against *ex vivo* juvenile worms, compounds **1-5** have superior activity to PZQ.

5) Although these inhibitors show some effect against juvenile worms, the reduction of egg burden does not seem to be better than that of PZQ. Therefore, statements in the abstract and discussion are not convincing in comparison with PZQ.

We amended the Discussion section to address these concerns.

Generally, in humans and in pre-clinical studies in mice, PZQ has either no or very moderate activity against juvenile worms even if it is used at very high 1000 mg/kg dose. It should be noted that in the animal models of juvenile infection, after treatment of juvenile worms, the mice are kept until the remaining worms mature (see Supplementary Methods). The mature worms do produce eggs.

In vivo activity. We tested compounds at day 23 p.i. Treatment with compound **2** at 1 X 100 mg/kg resulted in >60% reduction in worm burdens. In previous studies, treatments with PZQ at 1 X 500 mg/kg resulted in 0% reductions at 21 and 28 days p.i. and treatments at 1 X 1000 mg/kg resulted in 50 % and 17% at day 21 and 28 p.i., respectively. In reference 43 (PMID: 15013742), treatments targeting 28 day old worms resulted in ED₅₀ = 2456 mg/kg. The ED₅₀ for compound **2** is < 100 mg/kg. On this basis, we conclude that compound **2** has superior *in vivo* activity to PZQ.

Text has been changed to focus on compound 2 in *in vivo* studies.

Title

It is rather difficult for readers who don't know TGR well to understand why the authors mention "non-covalent". Please explain the reason in the text more.

The Introduction section has been expanded to address this concern.

Line 145: The decrease in GSH/GSSG ratio does not correlate with the inhibition potency of GTR. This leads to questions if the target/effect is really correlated to GTR or is caused by a non-specific effect due to the high concentration of the compounds tested in these experiments (50 μ M).

In worms the GSH/GSSG ratio is maintained by TGR alone as worms do not have a GR enzyme. Decreases in GSH/GSSG ratio result from TGR inhibition. Treatments with PZQ and oltipraz, which both kill worms at the tested concentrations, have no effect on GSH/GSSG. This indicates that the GSH/GSSG in dying worms (PZQ/oltipraz treatments) is unaffected – we know that neither PZQ nor oltipraz inhibit TGR.

The fact that the compounds have greater activity against TGR in worms vs. recombinant TGR can be explained by many differences existing between *in vivo* and *in vitro* assays. For instance, the concentration of TGR in worms is not known and may be much lower than the 4 nM used in biochemical assays; the concentration of NADPH in worms is not known and may be different from the saturating concentrations (100 μ M) used in biochemical assays; the conformation of TGR in worms (reflective of TGR redox status) is not known and maybe different from that of TGR in a test tube. Although the concentrations of compounds used in this study were high, exposure to the compounds was brief, only 3 hr, and a significant reduction in GSH/GSSG was seen after only 1 hr exposure.

TGR inhibition has been validated in two orthogonal assays measuring GSH/GSSG ratio and TRFS green probe measuring activity of TGR. We provide an extensive discussion addressing this concern in the Discussion section.

Line 158: It is recommended to describe that a selective fluorescent probe can work for imaging the enzyme activity.

Details of TRFS-Green activity and specificity are given in reference 33 (PMID: 24351040) cited in the text. To clarify, text has been added to describe this method in the recommended section.

Line 162: Fig. 2a

The experiment was done at 5 μ M conc of compounds, which is a low concentration of the compounds where TGR is not inhibited.

Cpd 1, 42.1% inhibition at 67 μ M;

Cpd 2, 28.7% inhibition at 67 μ M;

Cpd 4, 68.6% inhibition at 67 μ M;

Cpd 7, IC₅₀ = 14.6 μ M;

Cpd 8, IC₅₀ = 10.3 μ M;

Why a decrease in fluorescence was observed at a concentration (5 μ M) much lower than the concentration required to observe some inhibition to TGR according to the data from table 1?

The compounds were tested at 30 μ M (explained in methods) not 5 μ M as reported in the figure. This has been corrected. Please see response to comments about GSH/GSSG Line 145 above for details concerning compound activity *in vivo* and *in vitro*.

Fig. 2c

The experiment was done at 30 μ M conc of the compounds.

Compounds were used at 30 μ M, 15 μ M, and 5 μ M. Please see response to comments about GSH/GSSG Line 145 above for details.

Line 164

From the extended data Fig 2a, b, c, it's hard to reach the conclusion that the novel TGR inhibitors engage TGR in ex vivo worms.

TRFS-Green is a highly selective TrxR and TGR substrate (DOI: 10.1021/ja408792k, ref # 33). It is nonfluorescent until converted by TGR to a fluorescent product. Inhibition of TGR prevents formation of the fluorescent product. The data show that fluorescence in worms is decreased in concentration- (panel c) and time- (panel a) dependent fashions on exposure to TGR inhibitors, with auranofin as a positive control. The untreated worms (--) show an increase in fluorescence, while all concentrations of inhibitor against treated worms have reduced fluorescence. Compound 12 is a negative control, structurally similar to active compounds, but a poor TGR inhibitor. Fluorescence in compound 12-treated worms is not significantly reduced compared to the control, untreated worms.

Line 174:

Please explain "the jump dilution assay" briefly to the readers.

A brief description has been added.

Although the new TGR inhibitors found by the authors are reversible, these inhibitors lack selectivity as they also inhibit the mammalian TrxR1 and raise questions about the safety profile.

This has been addressed in our answer to item 1 Rev #4 above.

Line 194

Why the inhibition of TGR is NADPH dependent? The cryo-EM structures of TGR with the non-covalent inhibitors were determined even in the absence of NADPH. If the non-covalent

inhibitors cannot bind to the TGR in absence of NADPH, does this mean that the cryo-EM structures described by the authors represent non-physiological structures?

The inhibition of TGR is NADPH-dependent because (i) TSAs show that most of the inhibitors specifically change the T_m of the NADP^+ -TGR(H) reduced species, (ii) inhibitors exert their action after incubation of the enzyme with NADPH and (iii) they show an uncompetitive mechanism of action. We believe that the destabilized NADP^+ -TGR(H) forms upon NADPH reduction (TGR has a lower T_m in this condition) can better accommodate the uncompetitive inhibitors having access to conformational states more suitable for compound binding (see Figure 5). However, not for all compounds the inhibition is strictly NADPH-dependent. The cryo-EM structure of TGR has been solved with the non-competitive inhibitor **9** able to bind both the free oxidized enzyme (E) and the NADP^+ -TGR(H) reduced species. Thus, in this case the cryo-EM structure describes a physiological state of the TGR-**9** complex. Compound **9** has been synthesized *ad hoc* to facilitate structural studies, replacing the hydrophobic cycloheptane ring of the uncompetitive inhibitor **8** with a sugar moiety. As now stated in the Results section “TGR inhibitors target a reduced form of TGR”, the change from uncompetitive to non-competitive mechanism of inhibitors, retaining the same binding site, is not rare in drug-design studies (doi: 10.1021/jm800512z; doi: 10.1002/cbic.201500119; doi: 10.1016/j.bioorg.2016.01.008). However, further studies on the mechanism of action of these compounds are ongoing.

Line 206:

K, V and T of T_m should be written in italic.

Corrected.

Line 210:

"poor" is not scientific word. I recommend editing the following descriptions used in other paragraphs, such as "our compound" or "better".

These words have been eliminated from the text.

Line 219:

At 500 μM of NADPH, can the reduced form of the enzyme also bind to NADPH, similar to what has been observed to the yeast type 2 NADH dehydrogenases (in this case NADH binds to the reduced form of the enzyme)?

Yes. The reduced form of TGR/TrxRs can bind another NADPH. In these oxidoreductases, 2 equivalents of NADPH are necessary to form the EH_4 species which is the one competent for substrate reduction. During turn-over TGR oscillates between the EH_2 and EH_4 species. Please, see new Figures 1a and 5c and the Introduction. Moreover, the electron transfer from NADPH/NADH to FAD is similar between TGRs/TrxRs and yeast type 2 NADH dehydrogenases as the modality of binding of the reducing substrate at the *re*-face of the FAD cofactors is in general maintained in both the enzyme families. (doi: 10.1038/nature11541).

Also, in the PRP labelling, there is labelling in the TGR in the absence of NADPH, although at a less effective rate.

The irreversible PRP has a lower propensity to bind the oxidized enzyme with respect to the reduced enzyme, in line with the uncompetitive behavior of the compounds from which its structure has been derived. However, the possibility of uncompetitive inhibitors binding a small/negligible fraction the free enzyme is an actual one, as true uncompetitive inhibitors that bind exclusively to the ES complex (or the downstream species) are extremely rare (Bellelli and Carey, Reversible Ligand Binding: Theory and Experiment, 2018).

Therefore, would you explain why the authors believe that these inhibitors bind to the ES complex and not to the reduced form of the enzyme?

We now better clarified this point in the Introduction and in the Results sections. We now refer to the functional reduced forms of the enzyme, EH_2 and EH_4 , as NADP^+ -TGR(H) complexes. The choice has been dictated by our previous observations that (i) the electron transfer between NADPH and the enzyme is fast and practically irreversible and (ii) both these reducing forms can retain a NADP^+ molecule (see ref. 17). We know that these bimolecular species (NADP^+ - EH_2 and NADP^+ - EH_4) are likely populated during the reductive half-reaction in TGRs/TrxRs, as they were isolated by X-ray crystallography by us (PDB ID: 2X99; ref. 17) and others (PDB ID: 1H&V). By structural and functional studies presented in the present manuscript, we propose that the compounds bound to the NADP^+ - EH_2 hamper NADP^+ release from the site, slowing down enzyme oscillations between EH_2 and EH_4 necessary for the turn-over (as shown in new Figure 5).

Why the compound 9 (a noncompetitive inhibitor and thus capable of binding the enzyme in the absence of NADP(H)) was not included in the TSA?

TSA data for compound **9** have been added to Figure 3. Compound **9** was designed for increased solubility for use in structural studies and it's a weaker binder with respect to the other inhibitors. Despite its non-competitive behavior observed in steady state experiments, compound **9** behaves as the other uncompetitive inhibitors: **9** doesn't shift the T_m of the oxidized enzyme but shifts that of the reduced one of + 1.5 °C at 400uM (~5 X IC_{50} , not shown) and of + 3 C° at 1mM (~14 X IC_{50} , now in Fig. 3). However, absence of a shift of a T_m in TSA doesn't necessarily imply that an actual binding is not present (PMID: 25630461; 26259992).

Line 242:

The authors explain the result of low solubility not only here but also later part. How have the authors tried the suitable conditions? More information on the trial should be described.

Before use, all the compounds were subjected to sonication-heat cycles to maximize their solubility in assays. Indication of their solubility was obtained by measuring the light scattering at the UV-Vis spectrophotometer of compound solutions at different concentrations, followed by centrifugation of the mixture and visual inspection. A description of this procedure has been added to the Methods section.

Line 278:

A description "generally" is not clear.

This portion of the title was removed.

Line 284:

Result on the enzyme from the Vero cell and the effect of the compounds on human cells should be described.

The TrxR enzyme tested was recombinant human TrxR1. The TrxR1 enzyme in Vero cells (*Chlorocebus* sp., XP_008002673.2) is 99% identical to human TrxR1 (494 of 499 identical amino acids). Inhibition of *Chlorocebus* TrxR1 and human TrxR1 is expected to be identical. Vero cells are commonly used in drug development studies in place of human cells.

Line 287:

Why do the authors describe the effect of cpd 6 on TrxR1 as "no appreciable inhibition" whilst as "significant inhibition" of cpd 2 on TGR, even if the degree of inhibition at 67 μ M for cpd 6/TrxR1 (28.9%) pairs and cpd 2/TGR (28.7%) are very close?

We deleted the statement on line 287 referred to. However, compounds **2** and **6** have a different inhibition behavior. Compound **6** is a fast inhibitor for both enzymes, reaching the equilibrium within 15 mins of preincubation (see Figure 3), allowing us to determine its selectivity versus TGR as we can measure a significant IC₅₀ for this enzyme (2.5 μ M); while, in case of TrxR, this was not possible as at 66.7 μ M the measured inhibition by **6** was below 50%, requiring high concentrations of the inhibitor at the limits of its solubility for IC₅₀ determination. Even though we can just provide an estimate of the IC₅₀ of compound **6** for TrxR (IC₅₀ > 66.7 μ M), we can safely state that this compound is at least 25X more potent against TGR than human TrxR. The differential activity of compound **6** TGR/hTrxR is now highlighted in the results and in the discussion sections.

Compound **2** is a slow inhibitor for TGR and a fast inhibitor for TrxR. It reaches the equilibrium with TGR after hours. Since, in these cases, it is not possible to provide good estimates of equilibrium parameters (as the apparent IC₅₀ changes with time), for TGR we determined the percentage of inhibition of compound **6** at 6 hours that cannot be compared with the percentage of inhibition measured at the equilibrium for TrxR. We now divided the previous Table 1 in two, clarifying these aspects.

Line 292:

Again, at such low concentrations, it doesn't seem to inhibit TGR.

As we explained in the text of the manuscript, the assays used to measure activity of TGR, TrxR, and related enzymes rely on unrealistically high concentrations of NADPH necessary for the assay. In the more physiological GSH/GSSG and TRFS green assays, these compounds clearly inhibit TGR because it is the only enzyme that can affect the GSH/GSSG ratio or generate the green fluorescent product from the TRFS-Green dye.

As mentioned before, "better" is not a scientific word.

Corrected

Line 302:

To what extent the structures of TGR from Sm and Sj are similar or different?

Can the effect on Sj be explained by, for example, its TGR modelled structure (Alpha fold model or any other model with high confidence)?

There is already an experimental structure of SjTGR solved by X-ray crystallography at a resolution of 2.3Å (PDB ID: 4LA1). With SmTGR, SjTGR shares residue identities of 91% overall and of 100% considering the residues that contribute the doorstep pocket (Please, see Supplementary Note K). The two structures are overall very similar (rmsd = 0.8 Å over 585 residues). This is in line with the similar effect observed for the compounds against *ex vivo* *S. japonicum* and *S. mansoni* as now stated in the Discussion Section.

Line 305:

"good" is not a scientific word.

It has been replaced with “significant”.

From lines 320 ~333:

This doesn't seem to be potent. Please include the data or compare them with the positive control group treated with PZQ.

The efficacy of compound **2** is 77% against adult worms in a single dose. This is certainly sufficiently high for a first iteration of compounds with a novel target that is different from that of PZQ. We are not trying to outcompete PZQ in efficacy but rather create a schistosomicidal agent with a novel target. Availability of multiple drugs acting at different targets is the key current strategy in combating infections and preventing the development of resistance.

From lines 334 ~339:

The activity against juveniles doesn't seem to be potent as well.

Why the authors did not investigate the effect of these compounds against schistosomula recovered from the lungs, as most of them seem to be more active to the NTS than in adults, as expected from Table 1.

The stage against which PZQ is least active is the liver stage, necessitating potent activity against this stage by potential new drugs. PZQ has similar activity against NTS and lung stage worms. A single dose of compound **2** is much more efficacious against *in vivo* liver juvenile worms than that reported for PZQ in the literature in mouse models of schistosomiasis. We report an ED₅₀ of <100 mg/kg for **2**, while for PZQ it is 2456 mg/kg. In general, the compounds were less active against *ex vivo* liver stage worms than against adults, with **1 – 5** having roughly half the activity against juvenile worms than adults. However, **1 – 5** are more active than PZQ against this stage. It has been reported that juvenile worms have higher levels of drug transporters than adults,

which may explain the decrease in activity. The goal of future studies will be to design efficacious TGR inhibitors that are not substrates for membrane transporters.

Line 350:

Will cpd 1 inhibit TGR at this plasma concentration (4.1 μ M)?

Yes, at 5 μ M, production of green fluorescent dye was abolished in TRFS-green assays (see corresponding sections in the main manuscript and Supplementary Information) Also, see our answers to similar questions concerning inhibition of TGR, GSH/GSSG ratio, TRFS-Green, and limitations of *ex vivo* LD₅₀ assays above. We expanded the Discussion section to address this concern.

Line 353:

Will cpd 2 inhibit TGR at this plasma concentration (4.8, 3.9, and 3.0 μ M)?

Yes, at 5 μ M, production of green fluorescent dye was abolished in TRFS-Green assays (see corresponding sections in the main manuscript and Supplementary Information) Also, see our answers to similar questions concerning inhibition of TGR, GSH/GSSG ratio, TRFS green, and limitations of *ex vivo* LD₅₀ assays above. We expanded the Discussion section to address this concern.

Line 356:

The toxicity study result should be shown.

“Toxicity” was replaced with “visible adverse effects” as no additional toxicity measurements were done.

Discussion

Line 440 ~443:

From the data presented by the authors, the in vivo effect of these TGR inhibitors does not appear to surpass the efficacy of PZQ to adult and egg burdens.

This aspect was addressed in our answers above.

Although these inhibitors have some effect against juvenile worms, the reduction of egg burden does not seem to be better than that of PZQ. Therefore, such a claim does not seem to be convincing when compared with PZQ.

This aspect was addressed in our answers above.

Line 454~457

Can the lack of specificity of the presented inhibitors between TGR and human TrxR1 be explained by their structures (determined or modelled)?

The sequence and structural alignment between TGR and hTrxR1 (for sequence alignment see Supplementary Note K) indicates that they do have substantial homology, but with differences in

their respective doorstep pockets. hTrxR1 displays 74% sequence identity in the doorstep pocket residues with respect to SmTGR. Remarkably, the charge distribution and shape of TrxR1 in this site is different with respect to SmTGR (please see ref. 33, Silvestri, I. et al. Fragment-Based Discovery of a Regulatory Site in Thioredoxin Glutathione Reductase Acting as "Doorstop" for NADPH Entry. ACS Chem. Biol. 13, 2190-2202 (2018).

<https://doi.org/10.1021/acscchembio.8b00349>) due to the presence of charged and bulky residues, i.e., E337, D338, E341, E368 and K389 in hTrxR in place of A436, G437, Q440, S467 and D488 in TGR. This indicates that selectivity can be attainable as is demonstrated by the differential activity of compound **6** against the two enzymes (see the Discussion section).

We are planning to address this very important aspect of selectivity in future optimization of these unoptimized lead compounds, also collecting high resolution cryo-EM data on different inhibitors in complex with TGR.

REVIEWERS' COMMENTS

Reviewer #1 (Remarks to the Author):

NCOMMS-22-44026A

Some of my concerns have been addressed in this revision, but others remain and require further clarification:

1) Returning to my original comments (3 and 4) concerning worm burden variabilities and their influence on how power calculations were derived to inform group sizes during efficacy studies, I would like the authors to explain how a group size of 3 animals (Supplementary Data I, panel a; compound 2, 1 X 100 mg/kg) meets their criteria (experimental power of 80% is the $1-\beta$, not α , value; risk of detecting a false negative result is 20%) when worm burdens (in this study alone) range from 11-38/mouse (these values and others collected by the authors through years of working on schistosomiasis would allow an effect size to be calculated; the authors have not mentioned this in their methodology; it is a key component for power calculations)?

If this number (3) of animals/group was sufficient for determining a significant result mediated by treatment, then why were larger group sizes ($n=5$) used in all other experiments (including the control group in the experiment in question). I ask because the experiment in question shows the largest effect and is the reason for much of the discussion around these analogues' in vivo efficacy. This particular dataset is representative of three replicates (mentioned in the Figure legend); according to ARRIVE guidelines (which the authors have followed according to the Nature Editorial Policy Checklist), the authors need to show all of the data (including the other two replicates). The inclusion of all data (for all compounds delivered in vivo) would more strongly support the authors' conclusions on the in vivo efficacy of this particular compound (compound 2).

As I previously have indicated, the in vivo efficacy experiments require greater detail and explanation. As they are presented, they do not fully meet the ARRIVE guidelines.

2) Comparisons to PZQ still require re-writing (original comment 7). While a direct ex vivo comparison has been performed for PZQ and the author's compounds (which are not being questioned), similar in vivo comparisons have not been completed. Therefore, broadly-sweeping statements such as those found in lines 338-340 and 448-450 need to be tempered as the authors are comparing their in vivo generated results to those derived in the literature from other laboratories assessing PZQ in vivo activity. Results are presented that the authors' compounds are active when delivered to infected mice

at 23 days post-infection (although the methods state 3 week so harmonisation for precision is necessary), but as they never directly compared to PZQ treated 23-day infected mice in parallel, some of their enthusiastic statements need tempering.

3) Table 1 and Table 2 are now split from original Table 1. However, the footnotes require editing as there seems to be remnants from original Table 1 in both OR insufficient details. For example, there is no 'e' in Table 1; likewise, there is no definition for 'e' in Table 2. Also, please define the abbreviations of AF, PZQ and MZM in these tables (as another footnote). In Table 2, please indicate that the values are LD50s as this is not indicated.

Reviewer #2 (Remarks to the Author):

The authors have suitably addressed my comments in their revision.

Reviewer #3 (Remarks to the Author):

The authors have suitably addressed my prior comments.

Reviewer #4 (Remarks to the Author):

The manuscript is much improved and most of our concerns are clarified. I will point out a point that the authors need to edit.

As a response to our comment #1, the author describes "The human redox network is redundant, with separate Trx and GSH systems with redundancy and compensatory activities so that inhibition of one arm can be supplemented by the other.

The explanation shown above should be discussed in the manuscript with appropriate citation(s).

We would like to thank the reviewers for reading the manuscript and their valuable suggestions. The reviewers comments to be addressed are in "italic". Our responses are in regular font in blue.

Revision 3:

Reviewer #1 (Remarks to the Author):

NCOMMS-22-44026A

Some of my concerns have been addressed in this revision, but others remain and require further clarification:

1) Returning to my original comments (3 and 4) concerning worm burden variabilities and their influence on how power calculations were derived to inform group sizes during efficacy studies, I would like the authors to explain how a group size of 3 animals (Supplementary Data I, panel a; compound 2, 1 X 100 mg/kg) meets their criteria (experimental power of 80% is the $1-\beta$, not α , value; risk of detecting a false negative result is 20%) when worm burdens (in this study alone) range from 11-38/mouse (these values and others collected by the authors through years of working on schistosomiasis would allow an effect size to be calculated; the authors have not mentioned this in their methodology; it is a key component for power calculations)?

If this number (3) of animals/group was sufficient for determining a significant result mediated by treatment, then why were larger group sizes ($n=5$) used in all other experiments (including the control group in the experiment in question). I ask because the experiment in question shows the largest effect and is the reason for much of the discussion around these analogues' in vivo efficacy. This particular dataset is representative of three replicates (mentioned in the Figure legend); according to ARRIVE guidelines (which the authors have followed according to the Nature Editorial Policy Checklist), the authors need to show all of the data (including the other two replicates). The inclusion of all data (for all compounds delivered in vivo) would more strongly support the authors' conclusions on the in vivo efficacy of this particular compound (compound 2).

As I previously have indicated, the in vivo efficacy experiments require greater detail and explanation. As they are presented, they do not fully meet the ARRIVE guidelines.

Based on the reviewer's suggestion, we conducted further *in vivo* experiments using compound 2 in order to expand the treated group size. The additional results, which validate the previous findings, exhibit enhanced statistical robustness, reaching the ARRIVE guidelines, and are now illustrated in Figure 6 of the main text.

The statement that the dataset is a representative of three replicates in the referenced figure legend was an error and has been corrected. All results are shown in the manuscript.

2) Comparisons to PZQ still require re-writing (original comment 7). While a direct *ex vivo* comparison has been performed for PZQ and the author's compounds (which are not being questioned), similar *in vivo* comparisons have not been completed. Therefore, broadly-sweeping statements such as those found in lines 338-340 and 448-450 need to be tempered as the authors are comparing their *in vivo* generated results to those derived in the literature from other laboratories assessing PZQ *in vivo* activity. Results are presented that the authors' compounds are active when delivered to infected mice at 23 days post-infection (although the methods state 3 weeks so harmonisation for precision is necessary), but as they never directly compared to PZQ treated 23-day infected mice in parallel, some of their enthusiastic statements need tempering.

The statements in the abstract, in the result and discussion sections have been tempered and the methods for the *in vivo* experiments are now coherent with the main text.

3) Table 1 and Table 2 are now split from original Table 1. However, the footnotes require editing as there seems to be remnants from original Table 1 in both OR insufficient details. For example, there is no 'e' in Table 1; likewise, there is no definition for 'e' in Table 2. Also, please define the abbreviations of AF, PZQ and MZM in these tables (as another footnote). In Table 2, please indicate that the values are LD50s as this is not indicated.

The inconsistencies are now corrected in table legends.

Reviewer #4 (Remarks to the Author):

The manuscript is much improved and most of our concerns are clarified. I will point out a point that the authors need to edit. As a response to our comment #1, the author describes "The human redox network is redundant, with separate Trx and GSH systems with redundancy and compensatory activities so that inhibition of one arm can be supplemented by the other. The explanation shown above should be discussed in the manuscript with appropriate citation(s).

A sentence has been added to the discussion together with a proper reference.